



# The Physical and Biogeochemical Parameters along the Coastal Waters of Saudi Arabia during Field Surveys in Summer, 2021

Yasser O. Abualnaja[1*], Alexandra Pavlidou[2**], James H. Churchill[3], Ioannis Hatzianestis[2], Dimitris Velaoras[2], Harilaos Kontoyiannis[2], Vassilis P. Papadopoulos[2], Aristomenis P. Karageorgis[2], Georgia Assimakopoulou[2], Helen Kaberi[2], Theodoros Kannelopoulos[2], Constantine Parinos[2], Christina Zeri[2], Dionysios Ballas[2], Elli Pitta[2], Vassiliki Paraskevopoulou[4], Afroditi Androni[2], Styliani Chourdaki[2], Vassileia Fioraki[2], Stylianos Iliakis[2], Georgia Kabouri[2], Angeliki Konstantinopoulou[2], Georgios Krokos[2], Dimitra Papageorgiou[2], Alkiviadis Papageorgiou[2], Georgios Pappas[2], Elvira Plakidi[2], Eleni Rousselaki[2], Ioanna Stavrakaki[2], Eleni Tzempelikou[2], Panagiota Zachioti[2], Anthi Yfanti[4], Theodore Zoulias [2], Abdulah Al Amoudi[5], Yasser Alshehri [5], Ahmad Alharbi [5], Hammad Raji Al Sulami [5], Taha Boksmati [5], Rayan Mutwalli [5], Ibrahim Hoteit [6]

[1] Red Sea Research Center, King Abdullah University of Science and Technology, Thuwal, Jeddah 23955-6900, Saudi Arabia
[2] Hellenic Centre for Marine Research - HCMR, Institute of Oceanography, Mavro Lithari, 19013, Greece
[3] Department of Physical Oceanography, Woods Hole Oceanographic Institution, Woods Hole, MA 02543, USA
[4] National and Kapodistrian University of Athens, Department of Chemistry, Laboratory of Environmental Chemistry, Zografou 15784, Greece
[5] National Center for Environmental Compliance, Jeddah, Saudi Arabia
[6] Physical Sciences and Engineering Division, King Abdullah University of Science and Technology (KAUST), Thuwal, Saudi Arabia

*Corresponding authors:*

*\*Red Sea Research Center, King Abdullah University of Science and Technology - Yasser Abualnaja, e-mail: Yasser.abualnaja@kaust.edu.sa, Tel: +96654470622*

*\*\*Hellenic Centre for Marine Research, Institute of Oceanography – Alexandra Pavlidou, e-mail: aleka@hcmr.gr, Tel: +30 22910 76365, Fax: +30 22910 76347.*

**Abstract.** During the last decades, the coastal areas of the Kingdom of Saudi Arabia, on the Red Sea and the Arabian Gulf, have been subjected to intense economic and industrial growth. As a result, it may be expected that the overall environmental status of Saudi Arabian coastal marine waters has been affected by human activities. As a consequence, adequate management of the Saudi Arabian coastal zone requires an assessment of how the various pressures within this zone impact the quality of

seawater and sediments. To this end, environmental surveys were conducted over fifteen hotspot areas (areas subject to environmental pressures) in the Saudi Arabian coastal zone of the Red Sea and over three hotspot areas in the Saudi Arabian waters of the Arabian Gulf. The survey in the Red Sea, conducted in June/July 2021, acquired measurements from hotspot areas spanning most of the Saudi coastline, extending from near the Saudi–Jordanian border in the north to Al Shuqaiq and Jizan Economic City (close to the Saudi–Yemen border) in the south. The survey in the Arabian Gulf, carried out in September

2021, included the areas of Al Khobar, Dammam, and Ras Al Khair. The main objective of both cruises was to record the physical and biogeochemical parameters along the coastal waters of the Kingdom, tracing the dispersion of contaminants related to specific pressures. Taken together, these cruises constitute the first multidisciplinary and geographically comprehensive study of contaminants within the Saudi Arabian coastal waters and sediments. The measurements acquired revealed the influence of various anthropogenic pressures on the coastal marine environment of Saudi Arabia and also

highlighted a strong influence of hydrographic conditions on the distribution of biochemical properties in the Red Sea and the Arabian Gulf. The data can be accessed at: SEANOE. https://doi.org/10.17882/96463 (Abualnaja et al., 2023), whereas the details of the sampling stations at https://mcep.kaust.edu.sa/cruise-postings. The dataset includes the parameters shown in Tables 1(a,b) and 2(a).

## 1. Introduction

The Red Sea and the Arabian (Persian) Gulf are water bodies of importance to the Middle East and Northern Africa (MENA) region, and particularly to the Kingdom of Saudi Arabia, which has coastlines on both water bodies.

An elongated, marginal oceanic basin, the Red Sea is bordered by Northeast Africa and the Arabian Peninsula. Spanning almost 20° in latitude, the Red Sea is more than 2200 km long and is roughly 200

km wide on average. The bathymetry of the Red Sea is characterised by a deep axial trench, exceeding 3000 m in depth, bordered by shallow (100-200 m) and broad coastal shelf platforms that cover more than 40% of the basin (Rasul et al., 2015). The shallow coastal areas are particularly wide in the southern Red Sea, where they take the form of extended shallow banks riddled with coral reefs complexes. The coastal areas in the north are narrower than those to the south and reach depths of more than 200 m. The shallow



Gulf of Suez and the deeper and narrower Gulf of Aqaba are the natural extensions of the Red Sea to the north. The Red Sea communicates with the Mediterranean Sea through the Suez Canal in the Gulf of Suez; however, this water exchange is practically negligible, and the Red Sea relies on the Indian Ocean for its water renewal through the Bab-el-Mandeb Strait at its southernmost edge. In climatic terms, the Red Sea is divided into two distinct parts: the southern part, which is affected by the Arabian Sea monsoon, and the northern part, where a seasonal cycle of warm and cold periods prevails (Abualnaja et al., 2015; Viswanadhapalli et al., 2017). The elongated shape of the basin, the water exchange with the Gulf of Aden (Indian Ocean), and local atmospheric forcing regulate the water properties and general circulation of the Red Sea. Relatively fresh seawater enters the basin at its southern end, counterbalancing the water deficit produced by excessive evaporation coupled with negligible precipitation and terrestrial runoff. This fresher water from the south, with typical salinity values of 36–37.5, travels northwards through a complicated near-surface circulation (Sofianos and Johns, 2003; 2015; Yao et al., 2014; Zhan et al., 2014). The general northward surface circulation, which becomes stronger in winter, includes northward currents that flow near the east coast of the Red Sea and have been described previously as the Eastern Boundary Current (Sofianos and Jones, 2003). This boundary current affects all coastal regions, creating small eddies and bifurcating currents along natural barriers in the shallow areas near the coasts. During winter, the already hypersaline water in the Gulf of Aqaba and Suez Gulf, along with the northern part of the Red Sea, cools and sinks to the intermediate and deep layers of the water column, depending on its density (e.g., Sofianos and Johns, 2003; 2015; Papadopoulos et al., 2013; 2015; Zhai et al., 2015; Yao and Hoteit, 2018; Asfahani et al., 2020).

Within the Saudi Arabian coastal zone of the Red Sea, main drivers of pollution include: maritime transport, fisheries, aquaculture, oil, gas and energy infrastructures, coastal industry, coastal and maritime tourism, municipal and industrial discharge, and urban development (Schröder et al., 2021).

In contrast to the Red Sea, the Arabian Gulf is a shallow plateau with relatively smooth bottom topography, especially in its northern and western parts. It is a semi-enclosed oceanic basin that extends between the eastern part of the Arabian Peninsula and the mountainous coastline of Iran. The gulf exchanges water with the Gulf of Oman (Indian Ocean) through the Strait of Hormuz at its south-eastern



edge. The mean depth of the Arabian Gulf is 35 m, with its maximum depth of 120 m found close to the Strait of Hormuz. The physical properties of Arabian Gulf water are regulated by the water exchange with the Indian Ocean and the high evaporation rate. It can be considered as an idiosyncratic seawater basin, strongly influenced by its surrounding arid-to-hyper-arid areas. It is highly affected by dust storms, which transport large amounts of material and nutrients that influence the local ecosystems (Gherboudj and Ghedira, 2014). The gulf receives a small annual amount of precipitation, around 15 cm/year (Reynolds, 1993), and a river discharge estimated at 18 cm/year (Sheppard et al., 1992). However, damming of the major rivers in the region has resulted in substantial reductions in freshwater discharge (Sheppard et al., 2010). Evaporation is very high throughout the gulf, especially over the western part, and reaches approximately 200 cm/year, resulting in unusually high salinities compared to the open ocean (Johns et al., 2003). The Arabian Gulf features some of the highest temperatures and salinities observed in any marine water body worldwide (Sheppard et al., 2010).  It is noteworthy that the maximum salinities within the gulf exceed 45 and are found along the Saudi Arabian, Bahraini, and Qatari coastal areas.

The current environmental status of the Arabian Gulf is changing and is increasingly impacted by numerous adverse anthropogenic factors (Al Azhar et al., 2016). Notable among these factors is activity associated with the petroleum industry (Jones et al., 2008).  More than 50% of the world's oil and natural gas reserves are located within the gulf.  Other factors include desalination, reduced discharge from the two major rivers of the Tigris and the Euphrates (due to damming for the increased water demands), urbanisation and rapid industrial and residential development.  The combination of these factors has substantially altered the environmental status of the Arabian Gulf, especially along the Saudi coast (Vaughan et al., 2019), and have resulted in the Arabian Gulf being classified among the highest anthropogenically impacted regions in the world (Naser, 2013; 2014; 2015; Schröder et al., 2021).

Despite the economic and environmental importance of the Red Sea and Arabian Gulf to the MENA region, and the potential impact of anthropogenic activities on these water bodies, quality of marine environment of the Red Sea and the Arabian Gulf has not been extensively studied. Those studies that have been directed at the quality of the Saudi Arabian marine environment have largely been locally focused, for example on the marine area near Jeddah and the Jeddah lagoons. A number of studies have



focused on trace metals (Al Farawati et al., 2011; Yussef, 2015; Brima abd Albishri, 2017; Al Mur et al.,
2017; Ali et al., 2017; Fallatah et al., 2018; Al Mur 2020; Mannaa et al., 2021; Halawani et al., 2022; El
Zokm et al., 2022) and/or organic pollutants (PAHs) (El Maradny et al., 2023). Very few studies, also
related mainly to metals and organic contaminants are limited to local marine environments as Al-Lith
(Abu Zied and Harini, 2017; Buntan et al., 2020), Yambu (El Sorogy et al., 2023), Jazan (Ali Kanal et
al., 2020). The sporadic and geographically confined nature of contaminant measurements acquired in the
Red Sea highlight the importance of the data set obtained in this work, since it extends along the whole
Saudi Arabian coastline of the Red Sea.

Similarly, few studies of contaminants have been conducted in the Saudi Arabian waters of the Arabian
Gulf.  These deal mainly with metals and petroleum (Freije, 2015; Alharbi et al., 2017; Al Kahtany et al.,
2018; El Sorogy et al., 2018; Alharbi and El Sorogy, 2019;1017; Paparella et al., 2022; Amin and
Almahasheer, 2022; Alharbi et al., 2022; Alzahrani et al 2023; Al Kahtany et al., 2023; Sohaib et al.,
2023).

As part of the Vision 2030 for economic growth and development in the Kingdom of Saudi Arabia, the
Marine and Coastal Environment Protection (MCEP) Initiative for Saudi Arabia was established
(https://mcep.kaust.edu.sa/). The objective of this project, a collaboration between the National Center
for Environmental Compliance (NCEC) and King Abdullah University of Science and Technology
(KAUST), was to provide a national status quo assessment to set priorities for protection of the Kingdom's
coastal environment. The project was divided into 7 tiered tasks (https://mcep.kaust.edu.sa/).  Guided by
the findings of Task 4, Hotspot Analysis, KAUST's partner - the Hellenic Centre for Marine Research
(HCMR) - undertook field surveillance at a number of sites in the Red Sea and Arabian Gulf. The sites
were selected in consultation with the NCEC on the basis of the findings from Task 4
(https://mcep.kaust.edu.sa/) (Schröder et al., 2021). The field surveillance, Task-6, was designed to trace
the discrete sources of pollution in critical hotspot areas. These sources include: wastewater treatment
plants, desalination plants, ports, industry, petroleum platforms, aquaculture facilities (floating cages and
onshore operations), and urban development. A variety of hydrographic and chemical properties in the
water column and sediment were measured in each area.

Here, we present physical and biochemical measurements obtained during the two surveillance cruises, conducted in June and September 2021. Together, these cruises constitute the first multidisciplinary and geographically comprehensive survey of contaminants in Saudi Arabian coastal zone. Here, these data are used to described hydrographic conditions and the spatial variability of biochemical variables in the

Red Sea and the Arabian Gulf. In addition, the baseline values for various pollutants, chlorophyll-a (Chl-a), nutrients, and rarely measured essential oceanographic variables, such as dissolved organic carbon, are presented along with their spatial distributions.

## 2.  Data Provenance

The survey in the Red Sea was carried out from the R/V *AEGAEO* of the HCMR from 9 June to 6 July

2021, whereas the survey in the Arabian Gulf was carried out using small fishing boats from 17 to 22 September 2021. The cruise in the Red Sea was conducted over a north–south coastal transect from the area located in the northern part of the Gulf of Aqaba close to the Saudi Arabia–Jordan border (Hagl) to the Jizan Economic City area located in the southern region of the Red Sea (Fig. 1a). The hot spots areas surveyed, from north to south, were: an area near the Saudi-Jordanian border (Phosphate Terminal in the

Port of Aqaba in Jordan; cross-border pollution) and Haql (desalination, power and sewage-treatment plants; port activities), Magna (maritime traffic), Tabuk Fisheries (aquaculture activities), Duba (desalination plant), Al Wajh (port facilities; desalination plant), Red Sea Project Lagoon (north and west channels), Yanbu Cement Company (industrial discharge), Yanbu King Fahd Port (industrial and shipping center -, the largest in Saudi Arabia on the basis of oil and overall cargo volume; Schröder et al.,

2021), Jeddah Lagoon System (wastewater inputs), Jeddah Mena (port operations), Al Khurma (wastewater treatment plant), Al Lith (shrimp and fish farms), Al-Shuqaiq (desalination plant), and Jazan Economic City (expanding industrial facility). During the cruise, measurements of hydrographic and biochemical variables were acquired with traditional techniques (i.e., use of a CTD with companion sensors, and Acoustic Doppler Current Profiler [ADCP] and collection of water samples). The sampling

strategy was aimed at resolving the dispersion of contaminants related to environmental pressures impacting each area. For example, when sampling near desalination and wastewater treatment facilities,

we first conducted a CTD survey, acquiring data on temperature, salinity, dissolved oxygen (DO) and turbidity to identify the signal of the discharge from these facilities. Subsequently, water and sediment samples were obtained within the discharge signal suggested by the CTD data and at a point distant from the signal (reference site). Samples of water and sediment, as well as CTD data, were acquired from both R/V *AEGAEO* and *AEGAEO*'s support vessel (tender). At deeper (>10 m) stations, water samples and vertical CTD profiles were acquired using *AEGAEO*'s rosette sampling system, which consists of a CTD profiler with twelve 12-L Niskin bottles. Sediment samples at the deeper stations were acquired with the ship's box corer. At shallower stations, the sampling was carried out using the tender. At these shallower stations, sediment samples were acquired with a small grab sampler, and water samples were taken using a single Niskin bottle lowered from the tender. Water samples were taken at discrete depths, such as surface, 10 m, 20 m and near the bottom (roughly 0.5 m from the seabed), as well as from depths of particular interest (e.g., where effluent plumes were identified) if they did not match with the discrete depths.

The shipboard CTD system included a sensor suite for acquiring *in situ* data of salinity, temperature, DO, pH, specific conductivity, total dissolved solids, turbidity, Chl-a and total suspended matter. Onboard analysis was performed on the water samples for ammonium, DO, biological oxygen demand (BOD), sulphides, fluoride and total chlorine. The water samples were also subjected to onboard particle filtration and extraction of organic compounds.

The surveillance cruise in the Arabian Gulf covered the critical hotspot areas predefined by Schröder et al. (2021). Three main areas were sampled: Al Khobar, Dammam and Ras Al Khair. Unfortunately, rough sea conditions prevented surveying more areas, particularly in the northern part of the Saudi Arabian waters. These will hopefully be surveyed in future field surveillance work. The sampling entailed measurements of seawater properties using a portable CTD unit and the collection of seawater and sediment samples from which various types of contaminants and other properties were measured. Altogether, CTD data, as well as seawater and sediment samples, were obtained from 14 coastal stations (Fig. 1b). Similar to the Red Sea surveillance, the sampling strategy was aimed at resolving the dispersion of contaminants related to environmental pressures impacting each survey area. Sediment samples were acquired with a small grab sampler, and water samples were taken by individual Niskin bottles lowered



from the boat. Water samples were taken at discrete depths: surface, 10 m and near the bottom (roughly 0.5 m from the seabed). The portable CTD data included *in situ* measurements of temperature, conductivity and turbidity. On-board chemical analyses were not performed, these were not possible on the small boat employed. Nevertheless, all samples were treated and were subjected to extraction of organic compounds onboard immediately after collection. The water samples were subjected to particle

and Chl-a filtration at the ALS-Arabia laboratories

## Data Coverage and Parameters Measured:

Coverage: 17–29° N, 34° –42° E

Location name: Red Sea

Date start: 9 June 2021

Date end: 6 July 2021

**Table 1(a):** List of sampling sites of the *AEGAEO* cruise in the Red Sea as seen in the National Center for Environmental Compliance (NCEC) database (https://mcep.kaust.edu.sa/cruise-postings); List of sampling stations: location, depth and date (CTD data have been measured at all stations).

| Area | Station | Latitude_North (deg min) | Longitude_East (deg min) | Depth (m) | Date_June |
|---|---|---|---|---|---|
| **Cement Plant** | HB01 | 24 15.499 | 37 33.048 | 603 | 16 |
| | HB02 | 24 15.810 | 37 33.715 | 170 | 16 |
| **Yanbu** | KF9 | 23 54.663 | 38 16.905 | 41 | 17 |
| | KF8 | 23 55.278 | 38 15.550 | 30 | 17 |
| | KF7 | 23 55.860 | 38 14.136 | 47 | 17 |
| | KF5 | 23 55.197 | 38 11.796 | 104 | 17 |
| | KF4 | 23 56.255 | 38 12.655 | 40 | 17 |
| | KF6 | 23 57.383 | 38 11.590 | 29 | 17 |
| | KF1 | 23 56.963 | 38 13.530 | 16 | 17 |
| | KF3 | 23 56.500 | 38 12.865 | 38 | 17 |
| **Mena Jeddah** | JM6 | 21 27.697 | 39 06.323 | 44 | 18 |
| | JM5 | 21 27.775 | 39 08.548 | 32 | 18 |
| | JM3 | 21 27.935 | 39 09.204 | 26 | 18 |
| | JM2 | 21 28.673 | 39 09.618 | 15 | 18 |





| | JM1 | 21 27.327 | 39 09.940 | 12 | 18 |
|---|---|---|---|---|---|
| | JM4 | 21 27.220 | 39 09.318 | 15 | 18 |
| **Al Khumrah** | JS4 | 21 19.300 | 39 05.701 | 81 | 18 |
| | GH1 | 21 18.353 | 39 05.805 | 71 | 18 |
| | JS8 | 21 19.335 | 39 05.653 | 80 | 18 |
| | JS6 | 21 19.379 | 39 05.639 | 80 | 18 |
| | JS10 | 21 19.245 | 39 05.725 | 80 | 18 |
| | JS5 | 21 19.291 | 39 05.648 | 82 | 18 |
| | JS9 | 21 19.280 | 39 05.750 | 75 | 18 |
| | JS11 | 21 19.257 | 39 05.080 | 398 | 18 |
| | JS12 | 21 19.271 | 39 05.347 | 295 | 18 |
| | JS3 | 21 19.303 | 39 05.695 | 81 | 18 |
| | JS13 | 21 19.930 | 39 05.410 | 80 | 18 |
| **Lagoon** | L1 | 21 29.995 | 39 08.955 | 15 | 19 |
| | L2 (off intended position) | 21 29.665 | 39 09.123 | 10 | 19 |
| | L2 (on position) | 21 29.737 | 39 09.886 | 18 | 19 |
| | L3 | 21 29.697 | 39 09.660 | 16 | 19 |
| | L7 | 21 29.776 | 39 10.005 | 1.5 | 19 |
| | L8 | 21 29.612 | 39 10.078 | 14 | 19 |
| | L9 | 21 29.397 | 39 10.118 | 6 | 19 |
| | L10 | 21 29.222 | 39 10.304 | 5 | 19 |
| | L11 | 21 29.211 | 39 10.51 | 4 | 19 |
| | L12 | 21 29.140 | 39 10.704 | 5 | 19 |
| | L13 | 21 29.368 | 39 10.892 | 5 | 19 |
| | L14 | 21 29.50 | 39 11.008 | 4.5 | 19 |
| | NI1 | 21 28.965 | 39 10.667 | 4 | 19 |
| | NI2 | 21 28.876 | 39 10.621 | 2 | 19 |
| | NI3 | 21 29.286 | 39 10.330 | 3 | 19 |
| **Al Lith** | AL7 | 20 08.500 | 40 12.090 | 43 | 20 |
| | AL2 | 20 08.050 | 40 13.640 | 18 | 20 |
| | AL1 | 20 06.930 | 40 12.170 | 60 | 20 |
| | AL5 | 20 08.560 | 40 12.710 | 24 | 20 |
| | AL8 | 20 08.501 | 40 15.279 | 9.6 | 20 |
| | AL4 | 20 08.972 | 40 14.140 | 11 | 20 |
| | AL3 | 20 08.967 | 40 14.831 | 8.6 | 20 |



| | | | | | |
|---|---|---|---|---|---|
| **Jazan Economic City** | JZ1 | 17 21.610 | 42 15.590 | 15 | 22 |
| | JZ2 | 17 20.970 | 42 15.680 | 16 | 22 |
| | JZ3 | 17 20.270 | 42 15.840 | 15 | 22 |
| | JZ4 | 17 19.600 | 42 16.000 | 15 | 22 |
| | JZ5 | 17 18.950 | 42 16.090 | 14 | 22 |
| | JZ6 | 17 18.290 | 42 16.260 | 14 | 22 |
| | JZ7 | 17 17.610 | 42 16.400 | 10 | 22 |
| | JZ8 | 17 16.970 | 42 16.550 | 12 | 22 |
| | JZ9 | 17 16.290 | 42 16.710 | 15 | 22 |
| | JZ10 | 17 15.640 | 42 16.850 | 15 | 22 |
| | JZ11 | 17 15.000 | 42 17.000 | 12 | 22 |
| | JZ12 | 17 14.320 | 42 17.170 | 12 | 22 |
| | JZ13 | 17 13.700 | 42 17.300 | 15 | 22 |
| **Al Shuqaiq** | SH8N OR SH8 | 17 36.588 | 42 05.061 | 15 | 22 |
| | SH7 | 17 37.594 | 42 05.135 | 7 | 22 |
| | SH6 | 17 37.928 | 42 04.390 | 8 | 22 |
| | SH4N | 17 38.386 | 42 04.643 | 4 | 22 |
| | SH1N | 17 39.395 | 42 04.108 | 5 | 22 |
| | SH5 | 17 38.110 | 42 03.050 | 7 | 22 |
| | SH4 | 17 38.643 | 42 03.395 | 8.6 | 22 |
| **RSP West** | RSP3 | 25 27.020 | 36 42.210 | 17 | 27 |
| | RSP5 | 25 24.860 | 36 41.470 | 22 | 27 |
| | RSP6 | 25 25.370 | 36 46.470 | 20 | 27 |
| | RSP1 | 25 33.361 | 36 40.688 | 13 | 27 |
| | RSP2 | 25 29.181 | 36 40.776 | 10 | 27 |
| | RSP4 | 25 26.969 | 36 41.145 | 29 | 27 |
| **RSP North** | N5 | 25 55.920 | 36 37.850 | 20 | 27 |
| | N1 | 25 53.736 | 36 40.507 | 5 | 27 |
| | N2 | 25 53.492 | 36 39.258 | 11 | 27 |
| | N3 | 25 53.015 | 36 38.035 | 15 | 27 |
| | N4 | 25.51.758 | 36 37.147 | 7 | 27 |
| **Al Wajh** | AW2 | 26 13.190 | 36 27.240 | 52 | 28 |
| | AW3 | 26 13.710 | 36 26.890 | 185 | 28 |
| | AW4 | 26 12.740 | 36 27.510 | 50 | 28 |
| | AW1 | 26 13.479 | 36 27.652 | 9 | 28 |
| | AW5 | 26 12.990 | 36 26.850 | 150 | 28 |





| **Duba Desalination** | DBDS | 27 21.180 | 35 39.640 | 258 | 28 |
|---|---|---|---|---|---|
| **Magna** | MG | 28 24.070 | 34 44.130 | 80 | 29 |
| **Haql** | HQ1 | 29 17.600 | 34 55.670 | 78 | 29 |
| | HQ2 | 29 17.650 | 34 55.310 | 295 | 29 |
| | HQ3 | 29 17.070 | 34 55.540 | 81 | 29 |
| | HQ4 | 29 18.120 | 34 55.790 | 180 | 29 |
| | HQ5 | 29 21.120 | 34 56.700 | 270 | 29 |
| **Tabuk Fisheries** | TB3 | 27 45.650 | 35 25.670 | 50 | 30 |
| | TB4 | 27 45.100 | 35 25.890 | 100 | 30 |
| | TB1 | 27 46.255 | 35 25.508 | 26 | 30 |
| | TB2 | 27 46.089 | 35 25.775 | 21 | 30 |

**Table 1(b):** A list of the ADCP measurements.

| Station | Time (local) | Latitude_North (deg min) | Longitude_East (deg min) | Depth (m) | Date_June | ADCP type |
|---|---|---|---|---|---|---|
| HAB2 | 17:45 | 24 15.000 | 37 33.000 | 603 | 16 | SADCP |
| HAB2 | 18:47 | 24 15.862 | 37 33.636 | 160 | 16 | SADCP |
| KF9 | 03:18 | 23 54.663 | 38 16.905 | 44 | 17 | Portable |
| KF8 | 04:03 | 23 55.284 | 38 15.455 | 38 | 17 | Portable |
| KF7 | 05:05 | 23 55.880 | 38 14.153 | 49 | 17 | Portable |
| KF5 | 06:15 | 23 55.176 | 38 11.796 | 103 | 17 | SADCP |
| KF4 | 07:09 | 23 56.206 | 38 12.668 | 48 | 17 | Portable |
| KF6 | 08:10 | 23 57.383 | 38 11.600 | 29 | 17 | Portable |
| KF1 | 08:49 | 23 56.963 | 38 13.530 | 18 | 17 | Portable |
| JM6 | 09:24 | 21 27.656 | 38 06.310 | 50 | 18 | Portable |
| JM5 | 10:15 | 21   27.769 | 38 08.587 | 32 | 18 | Portable |
| JM3 | 10:50 | 21 27.934 | 39 09.209 | 27 | 18 | Portable |
| JM2 | 11:18 | 21 28.709 | 39 09.605 | 17 | 18 | Portable |
| JM1 | 11:55 | 21 27.315 | 39 09.894 | 14 | 18 | Portable |
| JM4 | 12:34 | 21 27.215 | 39 09.302 | 17 | 18 | Portable |
| JS4 | 15:05 | 21 19.300 | 39 05.711 | 81 | 18 | Portable |
| GH1 | 15:30 | 21 18.574 | 39 05.010 | 74 | 18 | Portable |





| JS8 | 15:55 | 21 19.546 | 39 05.659 | 82 | 18 | Portable |
|------|-------|-----------|-----------|-----|-----|----------|
| JS6 | 16:18 | 21 19.373 | 39 05.707 | 82 | 18 | Portable |
| JS10 | 16:42 | 21 19.240 | 39 05.738 | 80 | 18 | Portable |
| JS5 | 16:59 | 21 19.286 | 39 05.652 | 85 | 18 | Portable |
| JS9 | 17:23 | 21 19.214 | 39 05.746 | 82 | 18 | Portable |
| JS11 | 18:22 | 21 19.262 | 39 05.033 | 346 | 18 | SADCP |
| JS3 | 19:00 | 21 19.303 | 39 05.659 | 77 | 18 | SADCP |
| AL7 | 13:14 | 20 08.509 | 40 12.096 | 45 | 20 | Portable |
| AL2NEW | 14:08 | 20 08.068 | 40 13.610 | 20 | 20 | Portable |
| AL1 | 15:46 | 20 06.570 | 40 12.090 | 60 | 20 | SADCP |
| AL5 | 16:47 | 20 08.560 | 40 12.710 | 26 | 20 | Portable |
| JZ13 | 06:00 | 17 13.687 | 42 17.302 | 17 | 22 | Portable |
| JZ12 | 06:38 | 17 14.333 | 42 17.160 | 17 | 22 | Portable |
| JZ11 | 06:59 | 17 15.007 | 42 17.000 | 13 | 22 | Portable |
| JZ10 | 07:16 | 17 15.646 | 42 16.865 | 17 | 22 | Portable |
| JZ9 | 08:16 | 17 16.303 | 42 16.727 | 17 | 22 | Portable |
| JZ8 | 08:31 | 17 16.979 | 42 16.554 | 13 | 22 | Portable |
| JZ7 | 08:46 | 17 17.626 | 42 16.403 | 14 | 22 | Portable |
| JZ6 | 09:12 | 17 18.304 | 42 16.273 | 16 | 22 | Portable |
| JZ5 | 09:30 | 17 18.962 | 42 16.102 | 17 | 22 | Portable |
| JZ4 | 09:58 | 17 19.621 | 42 16.027 | 17 | 22 | Portable |
| JZ3 | 10:17 | 17 20.303 | 42 15.860 | 17 | 22 | Portable |
| JZ2 | 10:33 | 17 21.000 | 42 15.704 | 18 | 22 | Portable |
| JZ1 | 11:02 | 17 21.624 | 42 15.618 | 18 | 22 | Portable |
| SH5 | 19:50 | 17 38.116 | 42 03.049 | 15 | 22 | Portable |
| RSP5 | 06:43 | 25 24.851 | 36 41.497 | 28 | 27 | Portable |
| RSP6 | 07:44 | 25 25.380 | 36 46.471 | 22 | 27 | Portable |
| RSP3 | 08:52 | 25 27.022 | 36 42.238 | 19 | 27 | Portable |
| N5 | 16:37 | 25 55.936 | 36 37.880 | 22 | 27 | Portable |



| AW5 | 06:20 | 26 13.029 | 36 26.868 | 156 | 28 | SADCP |
|-----|-------|-----------|-----------|-----|----|-------|
| AW4 | 07:44 | 26 12.723 | 36 27.489 | 52 | 28 | Portable |
| AW2 | 08:30 | 26 13.261 | 36 27.266 | 50 | 28 | Portable |
| AW3 | 09:19 | 26 13.674 | 36 26.941 | 180 | 28 | SADCP |
| MG | 08:10 | 28 24.077 | 34 44.155 | 90 | 29 | SADCP |
| MG | 08:20 | 28 24.067 | 34 44.156 | 75 | 29 | Portable |
| HQ5 | 14:31 | 29 21.095 | 34 55.685 | 270 | 29 | SADCP |
| HQ4 | 15:45 | 29 18.075 | 34 55.475 | 180 | 29 | SADCP |
| HQ1 | 16:24 | 29 17.599 | 34 55.675 | 83 | 29 | Portable |
| HQ1 | 16:30 | 29 17.605 | 34 55.672 | 92 | 29 | SADCP |
| HQ3 | 16:58 | 29 17.083 | 34 55.540 | 86 | 29 | SADCP |
| HQ2 | 17:30 | 29 17.375 | 34 55.311 | 293 | 29 | SADCP |
| TB3 | 12:43 | 27 45.638 | 35 25.658 | 43 | 30 | Portable |
| TB4 | 13:30 | 27 45.092 | 35 25.897 | 114 | 30 | SADCP |





**Table 1(c):** List of biogeochemical parameters in the water column of the Red Sea cruise with R/V *AEGAEO* as seen in the NCEC database.

| Area | Station | DO and Nutrients | BOD | Fluoride | Sulphides | TOC | Chl-a | SPM | Chlorine | Cyanide | TPH | Oil and Grease | Chlorophenols | Metals |
|---|---|---|---|---|---|---|---|---|---|---|---|---|---|---|
| **Cement Plant** | HB01 | 2 | 2 | 2 | 2 | 2 | 2 | 2 | 2 | 2 | 2 | 2 | 2 | 2 |
| | | 20 | 20 | 20 | 20 | 20 | 20 | 20 | 20 | 20 | 20 | 20 | 20 | 20 |
| | | 50 | 50 | 50 | 50 | 50 | 50 | 50 | 50 | 50 | 50 | 50 | 50 | 50 |
| | HB02 | 2 | 2 | 2 | 2 | 2 | 2 | 2 | 2 | 2 | 2 | 2 | 2 | 2 |
| | | 20 | 20 | 20 | 20 | 20 | 20 | 20 | 20 | 20 | 20 | 20 | 20 | 20 |
| | | 50 | 50 | 50 | 50 | 50 | 50 | 50 | 50 | 50 | 50 | 50 | 50 | 50 |
| **Yanbu** | KF9 | 2 | 2 | 2 | 2 | 2 | 2 | 2 | 2 | 2 | 2 | 2 | 2 | 2 |
| | | 20 | 20 | 20 | 20 | 20 | 20 | 20 | 20 | 20 | 20 | 20 | 20 | 20 |
| | | 40 | 40 | 40 | 40 | 40 | 40 | 40 | 40 | 40 | 40 | 40 | 40 | 40 |
| | KF8 | 2 | 2 | 2 | 2 | 2 | 2 | 2 | 2 | 2 | 2 | 2 | 2 | 2 |
| | | 27 | 27 | 27 | 27 | 27 | 27 | 27 | 27 | 27 | 27 | 27 | 27 | 27 |
| | KF7 | 2 | 2 | 2 | 2 | 2 | 2 | 2 | 2 | 2 | 2 | 2 | 2 | 2 |
| | | 20 | 20 | 20 | 20 | 20 | 20 | 20 | 20 | 20 | 47 | 47 | 47 | 20 |
| | | 47 | 47 | 47 | 47 | 47 | 47 | 47 | 47 | 47 | | | | 47 |
| | KF5 | 2 | 2 | 2 | 2 | 2 | 2 | 2 | 2 | 2 | 2 | 2 | 2 | 2 |
| | | 20 | 20 | 20 | 20 | 20 | 20 | 20 | 20 | 20 | 20 | 20 | 20 | 20 |
| | | 104 | 104 | 104 | 104 | | | | | | | | | |
| | KF6 | 2 | 2 | 2 | 2 | 2 | 2 | 2 | 2 | 2 | 2 | 2 | 2 | 2 |
| | | 27 | 27 | 27 | 27 | 27 | 27 | 27 | 27 | 27 | 27 | 27 | 27 | 27 |
| | KF1 | 2 | 2 | 2 | 2 | 2 | 2 | 2 | 2 | 2 | 2 | 2 | 2 | 2 |
| | | 16 | 16 | 16 | 16 | 16 | 16 | 16 | 16 | 16 | 16 | 16 | 16 | 16 |
| | KF3 | 2 | 2 | 2 | 2 | 2 | 2 | 2 | 2 | 2 | 2 | 2 | 2 | 2 |
| | | 20 | 20 | 20 | 20 | 20 | 20 | 20 | 20 | 20 | 20 | 20 | 20 | 20 |
| | | 38 | 38 | 38 | 38 | 38 | 38 | 38 | 38 | 38 | 38 | 38 | 38 | 38 |
| **Mena Jeddah** | JM6 | 2 | 2 | 2 | 2 | 2 | 2 | 2 | 2 | 2 | 2 | 2 | 2 | 2 |
| | | 43 | 43 | 43 | 43 | 43 | 43 | 43 | 43 | 43 | 43 | 43 | 43 | 43 |
| | JM5 | 2 | 2 | 2 | 2 | 2 | 2 | 2 | 2 | 2 | 2 | 2 | 2 | 2 |
| | | 33 | 33 | 33 | 33 | 33 | 33 | 33 | 33 | 33 | 33 | 33 | 33 | 33 |
| | JM3 | 2 | 2 | 2 | 2 | 2 | 2 | 2 | 2 | 2 | 2 | 2 | 2 | 2 |
| | | 25 | 25 | 25 | 25 | 25 | 25 | 25 | 25 | 25 | 25 | 25 | 25 | 25 |



| Region | Station | | | | | | | | | | | | | |
|---|---|---|---|---|---|---|---|---|---|---|---|---|---|---|
| | JM2 | 2 | 2 | 2 | 2 | 2 | 2 | 2 | 2 | 2 | 2 | 2 | 2 | 2 |
| | JM1 | 2 | 2 | 2 | 2 | 2 | 2 | 2 | 2 | 2 | 2 | 2 | 2 | 2 |
| | | 12 | 12 | 12 | 12 | 12 | 12 | 12 | 12 | 12 | 12 | 12 | 12 | 12 |
| | JM4 | 2 | 2 | 2 | 2 | 2 | 2 | 2 | 2 | 2 | 2 | 2 | 2 | 2 |
| | | 15 | 15 | 15 | 15 | 15 | 15 | 15 | 15 | 15 | 15 | 15 | 15 | 15 |
| **Al Khumrah** | JS9 | 2 | 2 | 2 | 2 | 2 | 2 | 2 | 2 | 2 | 2 | 2 | 2 | 2 |
| | | 17 | 17 | 17 | 17 | 17 | 17 | 17 | 17 | 17 | 17 | 17 | 17 | 17 |
| | JS11 | 2 | 2 | 2 | 2 | 2 | 2 | 2 | 2 | 2 | 2 | 2 | 2 | 2 |
| | | 14 | 14 | 14 | 14 | 14 | 14 | 14 | 14 | 14 | 14 | 14 | 14 | 14 |
| | JS3 | 2 | 2 | 2 | 2 | 2 | 2 | 2 | 2 | 2 | 2 | 2 | 2 | 2 |
| | | 25 | 25 | 25 | 25 | 25 | 25 | 25 | 25 | 25 | 25 | 25 | 25 | 25 |
| | | 36 | 36 | 36 | 36 | 36 | 36 | 36 | 36 | 36 | 36 | 36 | 36 | 36 |
| **Lagoon** | L1 | 2 | 2 | 2 | 2 | 2 | 2 | 2 | 2 | 2 | 2 | 2 | 2 | 2 |
| | | | | | | | | | | 15 | 15 | 15 | 15 | |
| | L3 | 2 | 2 | 2 | 2 | 2 | 2 | 2 | 2 | 2 | 2 | 2 | 2 | 2 |
| | | 15 | 15 | 15 | 15 | 15 | 15 | 15 | 15 | 15 | 15 | 15 | 15 | 15 |
| | L9 | 2 | 2 | 2 | 2 | 2 | 2 | 2 | 2 | 2 | 2 | 2 | 2 | 2 |
| | | 6 | 6 | 6 | 6 | 6 | 6 | 6 | 6 | 6 | 6 | 6 | 6 | 6 |
| | L10 | 2 | 2 | 2 | 2 | 2 | 2 | 2 | 2 | 2 | 2 | 2 | 2 | 2 |
| | L12 | 2 | 2 | 2 | 2 | 2 | 2 | 2 | 2 | 2 | 2 | 2 | 2 | 2 |
| | L14 | 2 | 2 | 2 | 2 | 2 | 2 | 2 | 2 | 2 | 2 | 2 | 2 | 2 |
| | NI2 | 2 | 2 | 2 | 2 | 2 | 2 | 2 | 2 | 2 | 2 | 2 | 2 | 2 |
| **Al Lith** | AL7 | 2 | 2 | 2 | 2 | 2 | 2 | 2 | 2 | 2 | 2 | 2 | 2 | 2 |
| | | 23 | 23 | 23 | 23 | 23 | 23 | 23 | 23 | 23 | 43 | 43 | 43 | 23 |
| | | 30 | 30 | 30 | 30 | 30 | 30 | 30 | 30 | 30 | | | | 30 |
| | | 43 | 43 | 43 | 43 | 43 | 43 | 43 | 43 | 43 | | | | 43 |
| | AL2 | 2 | 2 | 2 | 2 | 2 | 2 | 2 | 2 | 2 | 2 | 2 | 2 | 2 |
| | | 18 | 18 | 18 | 18 | 18 | 18 | 18 | 18 | 18 | 18 | 18 | 18 | 18 |
| | AL1 | 2 | 2 | 2 | 2 | 2 | 2 | 2 | 2 | 2 | 2 | 2 | 2 | 2 |
| | | 23 | 23 | 23 | 23 | 23 | 23 | 23 | 23 | 23 | 61 | 61 | 61 | 23 |
| | | 61 | 61 | 61 | 61 | 61 | 61 | 61 | 61 | 61 | | | | 61 |
| | AL5 | 2 | 2 | 2 | 2 | 2 | 2 | 2 | 2 | 2 | 2 | 2 | 2 | 2 |
| | | 24 | 24 | 24 | 24 | 24 | 24 | 24 | 24 | 24 | 24 | 24 | 24 | 24 |
| | AL8 | 2 | 2 | 2 | 2 | 2 | 2 | 2 | 2 | 2 | 2 | 2 | 2 | 2 |
| | AL4 | 2 | 2 | 2 | 2 | 2 | 2 | 2 | 2 | 2 | 2 | 2 | 2 | 2 |
| | | 11 | 11 | 11 | 11 | 11 | 11 | 11 | 11 | 11 | 11 | 11 | 11 | 11 |
| | AL3 | 2 | 2 | 2 | 2 | 2 | 2 | 2 | 2 | 2 | 2 | 2 | 2 | 2 |
| **Jizan Econ City** | JZ2 | 2 | 2 | 2 | 2 | 2 | 2 | 2 | 2 | 2 | 2 | 2 | 2 | 2 |
| | | 17 | 17 | 17 | 17 | 17 | 17 | 17 | 17 | 17 | 17 | 17 | 17 | 17 |
| | JZ5 | 2 | 2 | 2 | 2 | 2 | 2 | 2 | 2 | 2 | 2 | 2 | 2 | 2 |
| | | 14 | 14 | 14 | 14 | 14 | 14 | 14 | 14 | 14 | 14 | 14 | 14 | 14 |




| Site | Station | | | | | | | | | | | | | |
|---|---|---|---|---|---|---|---|---|---|---|---|---|---|---|
| | JZ7 | 2 | 2 | 2 | 2 | 2 | 2 | 2 | 2 | 2 | 2 | 2 | 2 | 2 |
| | | 12 | 12 | 12 | 12 | 12 | 12 | 12 | 12 | 12 | 12 | 12 | 12 | 12 |
| | JZ10 | 2 | 2 | 2 | 2 | 2 | 2 | 2 | 2 | 2 | 2 | 2 | 2 | 2 |
| | | 14 | 14 | 14 | 14 | 14 | 14 | 14 | 14 | 14 | 14 | 14 | 14 | 14 |
| | JZ12 | 2 | 2 | 2 | 2 | 2 | 2 | 2 | 2 | 2 | 2 | 2 | 2 | 2 |
| | | 15 | 15 | 15 | 15 | 15 | 15 | 15 | 15 | 15 | 15 | 15 | 15 | 15 |
| | JZ13 | 2 | 2 | 2 | 2 | 2 | 2 | 2 | 2 | 2 | 2 | 2 | 2 | 2 |
| | | 15 | 15 | 15 | 15 | 15 | 15 | 15 | 15 | 15 | 15 | 15 | 15 | 15 |
| **Al Shuqaiq** | SH8 N | 2 | 2 | 2 | 2 | 2 | 2 | 2 | 2 | 2 | 2 | 2 | 2 | 2 |
| | | 14 | 14 | 14 | 14 | 14 | 14 | 14 | 14 | 14 | 14 | 14 | 14 | 14 |
| | SH6 | 2 | 2 | 2 | 2 | 2 | 2 | 2 | 2 | 2 | 2 | 2 | 2 | 2 |
| | | 5 | 5 | 5 | 5 | 5 | | | | | | | | 5 |
| | SH4N | 2 | 2 | 2 | 2 | 2 | 2 | 2 | 2 | 2 | 2 | 2 | 2 | 2 |
| | SH1N | 2 | 2 | 2 | 2 | 2 | 2 | 2 | 2 | 2 | 2 | 2 | 2 | 2 |
| | SH4 | 2 | 2 | 2 | 2 | 2 | 2 | 2 | 2 | 2 | 2 | 2 | 2 | 2 |
| | | 8 | 8 | 8 | 8 | 8 | 8 | 8 | 8 | 8 | 8 | 8 | 8 | 8 |
| **RSP** | RSP3 | 2 | 2 | 2 | 2 | 2 | 2 | 2 | 2 | 2 | 2 | 2 | 2 | 2 |
| | | 16 | 16 | 16 | 16 | 16 | 16 | 16 | 16 | 16 | 16 | 16 | 16 | 16 |
| | RSP5 | 2 | 2 | 2 | 2 | 2 | 2 | 2 | 2 | 2 | 2 | 2 | 2 | 2 |
| | | 10 | 10 | 10 | 10 | 10 | 10 | 10 | 10 | 10 | 10 | 10 | 10 | 10 |
| | | 21 | 21 | 21 | 21 | 21 | 21 | 21 | 21 | 21 | 21 | 21 | 21 | 21 |
| | RSP6 | 2 | 2 | 2 | 2 | 2 | 2 | 2 | 2 | 2 | 2 | 2 | 2 | 2 |
| | | 10 | 10 | 10 | 10 | 10 | 10 | 10 | 10 | 10 | 20 | 20 | 20 | 10 |
| | | 20 | 20 | 20 | 20 | 20 | 20 | 20 | 20 | 20 | | | | 20 |
| | RSP1 | 2 | 2 | 2 | 2 | 2 | 2 | 2 | 2 | 2 | 2 | 2 | 2 | 2 |
| | | 12 | 12 | 12 | 12 | 12 | 12 | | 12 | 12 | 12 | 12 | 12 | 12 |
| | RSP2 | 2 | 2 | 2 | 2 | 2 | 2 | 2 | 2 | 2 | 2 | 2 | 2 | 2 |
| | | 9 | 9 | 9 | 9 | 9 | 9 | | 9 | 9 | 9 | 9 | 9 | 9 |
| | RSP4 | 2 | 2 | 2 | 2 | 2 | 2 | 2 | 2 | 2 | 2 | 2 | 2 | 2 |
| | | 28 | 28 | 28 | 28 | 28 | 28 | | 28 | 28 | 28 | 28 | 28 | 28 |
| **RSP North** | N5 | 2 | 2 | 2 | 2 | 2 | 2 | 2 | 2 | 2 | 2 | 2 | 2 | 2 |
| | | 18 | 18 | 18 | 18 | 18 | 18 | 18 | 18 | 18 | 18 | 18 | 18 | 18 |
| | N1 | 2 | 2 | 2 | 2 | 2 | 2 | 2 | 2 | 2 | 2 | 2 | 2 | 2 |
| | N2 | 2 | 2 | 2 | 2 | 2 | 2 | 2 | 2 | 2 | 2 | 2 | 2 | 2 |
| | | 11 | 11 | 11 | 11 | 11 | 11 | 11 | 11 | 11 | 11 | 11 | 11 | 11 |
| | N3 | 2 | 2 | 2 | 2 | 2 | 2 | 2 | 2 | 2 | 2 | 2 | 2 | 2 |
| | | 15 | 15 | 15 | 15 | 15 | 15 | 15 | 15 | 15 | 15 | 15 | 15 | 15 |
| | N4 | 2 | 2 | 2 | 2 | 2 | 2 | 2 | 2 | 2 | 2 | 2 | 2 | 2 |
| **Al Wajh** | AW2 | 2 | 2 | 2 | 2 | 2 | 2 | 2 | 2 | 2 | 2 | 2 | 2 | 2 |
| | | 20 | 20 | 20 | 20 | 20 | 20 | 20 | 20 | 20 | 50 | 50 | 50 | 20 |
| | | 50 | 50 | 50 | 50 | 50 | 50 | 50 | 50 | 50 | | | | 50 |
| | AW3 | 2 | 2 | 2 | 2 | 2 | 2 | 2 | 2 | 2 | 2 | 2 | 2 | 2 |





| | | | | | | | | | | | | | | |
|---|---|---|---|---|---|---|---|---|---|---|---|---|---|---|
| | | 50 | 50 | 50 | 50 | 50 | 50 | 50 | 50 | 50 | 182 | 182 | 182 | 50 |
| | | 100 | 100 | 100 | 100 | 100 | 100 | 100 | 100 | 100 | | | | 100 |
| | | 182 | 182 | 182 | 182 | 182 | | 182 | 182 | 182 | | | | 182 |
| | AW4 | 2 | 2 | 2 | 2 | 2 | 2 | 2 | 2 | 2 | 2 | 2 | 2 | 2 |
| | | 30 | 30 | 30 | 30 | 30 | 30 | 30 | 30 | 30 | 47 | 47 | 47 | 30 |
| | | 47 | 47 | 47 | 47 | 47 | 47 | 47 | 47 | 47 | | | | 47 |
| | AW1 | 2 | 2 | 2 | 2 | 2 | 2 | 2 | 2 | 2 | 2 | 2 | 2 | 2 |
| | | 8 | 8 | 8 | 8 | 8 | 8 | 8 | 8 | 8 | 8 | 8 | 8 | 8 |
| | AW5 | 2 | 2 | 2 | 2 | 2 | 2 | 2 | 2 | 2 | 2 | 2 | 2 | 2 |
| | | 30 | 30 | 30 | 30 | 30 | 30 | 30 | 30 | 30 | 145 | 145 | 145 | 30 |
| | | 70 | 70 | 70 | 70 | 70 | 70 | 70 | 70 | 70 | | | | 70 |
| | | 145 | 145 | 145 | 145 | 145 | | 145 | 145 | 145 | | | | 145 |
| **Duba Desalination** | DBDS | 2 | 2 | 2 | 2 | 2 | 2 | 2 | 2 | 2 | 2 | 2 | 2 | 2 |
| | | 50 | 50 | 50 | 50 | 50 | 50 | 50 | 50 | 50 | 254 | 254 | 254 | 50 |
| | | 100 | 100 | 100 | 100 | 100 | 100 | 100 | 100 | 100 | | | | 100 |
| | | 254 | 254 | 254 | 254 | 254 | | 254 | 254 | 254 | | | | 254 |
| **Magna** | MG | 2 | 2 | 2 | 2 | 2 | 2 | 2 | 2 | 2 | 2 | 2 | 2 | 2 |
| | | 20 | 20 | 20 | 20 | 20 | 20 | 20 | 20 | 20 | 80 | 80 | 80 | 20 |
| | | 80 | 80 | 80 | 80 | 80 | 80 | 80 | 80 | 80 | | | | 80 |
| **Haql** | HQ1 | 2 | 2 | 2 | 2 | 2 | 2 | 2 | 2 | 2 | 2 | 2 | 2 | 2 |
| | | 78 | 78 | 78 | 78 | 78 | 78 | 78 | 78 | 78 | 78 | 78 | 78 | 78 |
| | HQ2 | 2 | 2 | 2 | 2 | 2 | 2 | 2 | | | | | | 2 |
| | | 100 | 100 | 100 | 100 | 100 | 100 | 100 | | | | | | 100 |
| | | 295 | 295 | 295 | 295 | 295 | | 295 | | | | | | 295 |
| | HQ3 | 2 | 2 | 2 | 2 | 2 | 2 | 2 | 2 | 2 | 2 | 2 | 2 | 2 |
| | | 81 | 81 | 81 | 81 | 81 | 81 | 81 | 81 | 81 | 81 | 81 | 81 | 81 |
| | HQ4 | 2 | 2 | 2 | 2 | 2 | 2 | 2 | 2 | 2 | 2 | 2 | 2 | 2 |
| | | 75 | 75 | 75 | 75 | 75 | 75 | 75 | 75 | 75 | 75 | 75 | 75 | 75 |
| | HQ5 | 2 | 2 | 2 | 2 | 2 | 2 | 2 | 2 | 2 | 2 | 2 | 2 | 2 |
| | | 20 | 20 | 20 | 20 | 20 | 20 | 20 | 20 | 20 | 270 | 270 | 270 | 20 |
| | | 100 | 100 | 100 | 100 | 100 | 100 | 100 | 100 | 100 | | | | 100 |
| | | 270 | 270 | 270 | 270 | 270 | | 270 | 270 | 270 | | | | 270 |
| **Tabuk Fisheries** | TB3 | 2 | 2 | 2 | 2 | 2 | 2 | 2 | 2 | 2 | 2 | 2 | 2 | 2 |
| | | 20 | 20 | 20 | 20 | 20 | 20 | 20 | 20 | 20 | 20 | 20 | 20 | 20 |
| | | 50 | 50 | 50 | 50 | 50 | 50 | 50 | 50 | 50 | 50 | 50 | 50 | 50 |
| | TB4 | 2 | 2 | 2 | 2 | 2 | 2 | 2 | 2 | 2 | 2 | 2 | 2 | 2 |
| | | 30 | 30 | 30 | 30 | 30 | 30 | 30 | 30 | 30 | 100 | 100 | 100 | 30 |
| | | 70 | 70 | 70 | 70 | 70 | 70 | 70 | 70 | 70 | | | | 70 |
| | | 100 | 100 | 100 | 100 | 100 | 100 | 100 | 100 | 100 | | | | 100 |
| | TB1 | 2 | 2 | 2 | 2 | 2 | 2 | 2 | 2 | 2 | 2 | 2 | 2 | 2 |
| | | 25 | 25 | 25 | 25 | 25 | 25 | 25 | 25 | 25 | 25 | 25 | 25 | 25 |
| | TB2 | 2 | 2 | 2 | 2 | 2 | 2 | 2 | 2 | 2 | 2 | 2 | 2 | 2 |





| | | 18 | 18 | 18 | 18 | 18 | 18 | 18 | 18 | 18 | 18 | 18 | 18 | 18 |
|---|---|---|---|---|---|---|---|---|---|---|---|---|---|---|

**Table 1(d):** List of sediment parameters from the cruise in the Red Sea with R/V *AEGAEO* as seen in the NCEC database.

| Area | Name | Date_ June | Metals_ Granulometry | Hydrocarbons | Cyanide | WS Cl | VOC | Phenols | PCB | Org. C/Carbonate | TN |
|---|---|---|---|---|---|---|---|---|---|---|---|
| **Yanbu** | KF9 | 17 | √ | √ | √ | √ | √ | √ | √ | √ | √ |
| | KF8 | 17 | √ | √ | √ | √ | √ | √ | √ | √ | √ |
| | KF7 | 17 | √ | √ | √ | √ | √ | √ | √ | √ | √ |
| | KF4 | 17 | √ | √ | √ | √ | √ | √ | √ | √ | √ |
| | KF6 | 17 | √ | √ | √ | √ | √ | √ | √ | √ | √ |
| | KF1 | 17 | √ | √ | √ | √ | √ | √ | √ | √ | √ |
| | KF3 | 17 | √ | √ | √ | √ | √ | √ | √ | √ | √ |
| **Mena Jeddah** | JM5 | 18 | √ | √ | √ | √ | √ | √ | √ | √ | √ |
| | JM3 | 18 | √ | √ | √ | √ | √ | √ | √ | √ | √ |
| | JM2 | 18 | √ | √ | √ | √ | √ | √ | √ | √ | √ |
| | JM1 | 18 | √ | √ | √ | √ | √ | √ | √ | √ | √ |
| **Al Khumrah** | JS3 | 18 | √ | √ | √ | √ | √ | √ | √ | √ | √ |
| **Lagoon** | L3 | 19 | √ | √ | √ | √ | | √ | √ | √ | √ |
| | L9 | 19 | √ | √ | √ | √ | √ | √ | √ | √ | √ |
| | L10 | 19 | √ | √ | √ | √ | | √ | √ | √ | √ |
| | L12 | 19 | √ | √ | √ | √ | | √ | √ | √ | √ |
| | L14 | 19 | √ | √ | √ | √ | √ | √ | √ | √ | √ |
| | NI2 | 19 | √ | √ | √ | √ | √ | √ | √ | √ | √ |
| **Al Lith** | AL7 | 20 | √ | √ | √ | √ | √ | √ | √ | √ | √ |
| | AL2 | 20 | √ | √ | √ | √ | √ | √ | √ | √ | √ |
| | AL1 | 20 | √ | √ | √ | √ | √ | √ | √ | √ | √ |
| | AL5 | 20 | √ | √ | √ | √ | √ | √ | √ | √ | √ |
| | AL4 | 20 | √ | √ | √ | √ | √ | √ | √ | √ | √ |
| | AL3 | 20 | √ | √ | √ | √ | √ | √ | √ | √ | √ |
| **Jazan Economic City** | JZ2 | 22 | √ | √ | √ | √ | √ | √ | √ | √ | √ |
| | JZ5 | 22 | √ | √ | √ | √ | √ | √ | √ | √ | √ |
| | JZ7 | 22 | √ | √ | √ | √ | √ | √ | √ | √ | √ |
| | JZ10 | 22 | √ | √ | √ | √ | √ | √ | √ | √ | √ |
| | JZ12 | 22 | √ | √ | √ | √ | √ | √ | √ | √ | √ |
| | JZ13 | 22 | √ | √ | √ | √ | √ | √ | √ | √ | √ |





| | | | | | | | | | | | |
|---|---|---|---|---|---|---|---|---|---|---|---|
| **Al Shuqaiq** | SH8N | 22 | √ | √ | √ | √ | √ | √ | √ | √ | √ |
| | SH6 | 22 | √ | √ | √ | √ | √ | √ | √ | √ | √ |
| | SH1N | 22 | √ | √ | √ | √ | √ | √ | √ | √ | √ |
| | SH5 | 22 | √ | | | | | | | √ | √ |
| | SH4 | 22 | √ | | | | | | | √ | √ |
| **RSP** | RSP3 | 27 | √ | √ | √ | √ | √ | √ | √ | √ | √ |
| | RSP2 | 27 | √ | √ | √ | √ | √ | √ | √ | √ | √ |
| | RSP4 | 27 | √ | √ | √ | √ | | √ | √ | √ | √ |
| **RSP North** | N5A | 27 | √ | √ | √ | √ | √ | √ | √ | √ | √ |
| | N1 | 27 | √ | √ | √ | √ | | √ | √ | √ | √ |
| | N2 | 27 | √ | √ | | | | √ | √ | √ | √ |
| **Al Wajh** | AW3 | 28 | √ | √ | √ | √ | √ | √ | √ | √ | √ |
| | AW4 | 28 | √ | √ | √ | √ | √ | √ | √ | √ | √ |
| | AW1 | 28 | √ | √ | √ | √ | √ | | √ | √ | √ |
| | AW5 | 28 | √ | √ | √ | √ | √ | √ | √ | √ | √ |
| **Duba Desalination** | TBDS | 28 | √ | √ | √ | √ | √ | √ | √ | √ | √ |
| **Magna** | MG | 29 | √ | √ | √ | √ | √ | √ | √ | √ | √ |
| **Haql** | HQ2 | 29 | √ | √ | √ | √ | √ | √ | √ | √ | √ |
| | HQ5 | 29 | √ | √ | √ | √ | √ | √ | √ | √ | √ |
| **Tabuk Fisheries** | TB3 | 30 | √ | √ | √ | √ | √ | √ | √ | √ | √ |
| | TB1 | 30 | √ | √ | √ | √ | √ | √ | √ | √ | √ |
| | TB2 | 30 | √ | √ | √ | √ | √ | √ | √ | √ | √ |





A link to the summary page of the Red Sea cruise can be found in the NCEC database under

https://mcep.kaust.edu.sa/cruise-postings.

Coverage: 26–27° N, 49° –50° E

Location name: Arabian Gulf

Date start: 17 September 2021

Date end: 22 September 2021


**Table 2(a):** List of sampling sites from the cruise in the Arabian Gulf; location, depth, date and sampling information (CTD data have been measured at all stations).

| Area | Latitude_North (deg min) | Longitude_East (deg min) | Name | Water Samples | Sediment Sample | Max Depth (m) | Samples Depth (m) |
|---|---|---|---|---|---|---|---|
| | | | | (S: near surface) | (Sed) | | |
| | | | | (M: mid-water) | | | |
| | | | | (B: near bottom) | | | |
| **Ras Al Khair** | 27 33.282 | 49 08.574 | RK1 | S | Sed | 5.3 | 2 |
| | 27 34.401 | 49 08.661 | RK3 | | Sed | 11 | |
| | 27 34.024 | 49 12.458 | RK4 | S | Sed | 17.6 | 2 |
| | 27 40.434 | 49 20.584 | RK7 | S | Sed | 13.5 | 2 |
| | 27 33.845 | 49 08.603 | RK2 | | | 9 | |
| **Dammam** | 26 29.186 | 50 08.911 | DM1 | S | Sed | 5.4 | 2 |
| | 26 31.080 | 50 09.289 | DM2 | | Sed | 6.5 | |
| | 26 32.109 | 50 09.738 | DM3 | S | Sed | 7.7 | 2 |
| | 26 43.807 | 50 17.341 | DM4 | S, B | Sed | 11 | 2, 10 |
| **Al Khobar** | 26 15.172 | 50 15.512 | RJ1 | S | Sed | 7.2 | 2 |
| | 26 13.640 | 50 14.847 | RJ2 | | | 8.2 | |
| | 26 12.315 | 50 14.310 | RJ3 | S | | 6.4 | |
| | 26 11.058 | 50 13.620 | RJ4 | S | | 7.2 | 5 |
| | 26 09.757 | 50 12.355 | RJ5 | | | 8.2 | |






**Table 2(b):** List of biogeochemical parameters in the water column (sampling depth) from the cruise in the Arabian Gulf.

| Area | Station | DO and Nutrients | Fluoride | TOC | Chl-a | SPM | Cyanide | TPH | Oil and Grease | Chlorophenols | Metals |
|---|---|---|---|---|---|---|---|---|---|---|---|
| **Ras Al Khair** | RK1 | 2 | 2 | 2 | 2 | 2 | 2 | 2 | 2 | 2 | 2 |
| | RK4 | 2 | 2 | 2 | 2 | 2 | 2 | 2 | 2 | 2 | 2 |
| | RK7 | 2 | 2 | 2 | 2 | 2 | 2 | 2 | 2 | 2 | 2 |
| **Dammam** | DM1 | 2 | 2 | 2 | 2 | 2 | 2 | 2 | 2 | 2 | 2 |
| | DM4 | 2 | 2 | 2 | 2 | 2 | 2 | 2 | 2 | 2 | 2 |
| | DM4 | 10 | 10 | 10 | 10 | 10 | 10 | 10 | 10 | 10 | 10 |
| **Al Khobar** | RJ1 | 2 | 2 | 2 | 2 | 2 | 2 | 2 | 2 | 2 | 2 |
| | RJ3 | 2 | 2 | 2 | 2 | 2 | 2 | 2 | 2 | 2 | 2 |
| | RJ4 | 5 | 5 | 5 | 5 | 5 | 5 | 5 | 5 | 5 | 5 |



**Table 2(c):** List of sediment parameters from the cruise in the Arabian Gulf.

| Area | Name | Metals Granulometry | Hydrocarbons | Cyanide | WS Cl | VOC | Phenols | PCB | Org. C/Carbonate | TN |
|---|---|---|---|---|---|---|---|---|---|---|
| **Ras Al Khair** | RK1 | √ | √ | √ | √ | √ | √ | √ | √ | √ |
| | RK3 | √ | √ | √ | √ | √ | √ | √ | √ | √ |
| | RK4 | √ | √ | √ | √ | √ | √ | √ | √ | √ |
| | RK7 | √ | √ | √ | √ | √ | √ | √ | √ | √ |
| **Dammam** | DM1 | √ | √ | √ | √ | √ | √ | √ | √ | √ |
| | DM2 | √ | √ | √ | √ | √ | √ | √ | √ | √ |
| | DM3 | √ | √ | √ | √ | √ | √ | √ | √ | √ |
| | DM4 | √ | √ | √ | √ | √ | √ | √ | √ | √ |
| **Al Khobar** | RJ1 | √ | √ | √ | √ | √ | √ | √ | √ | √ |

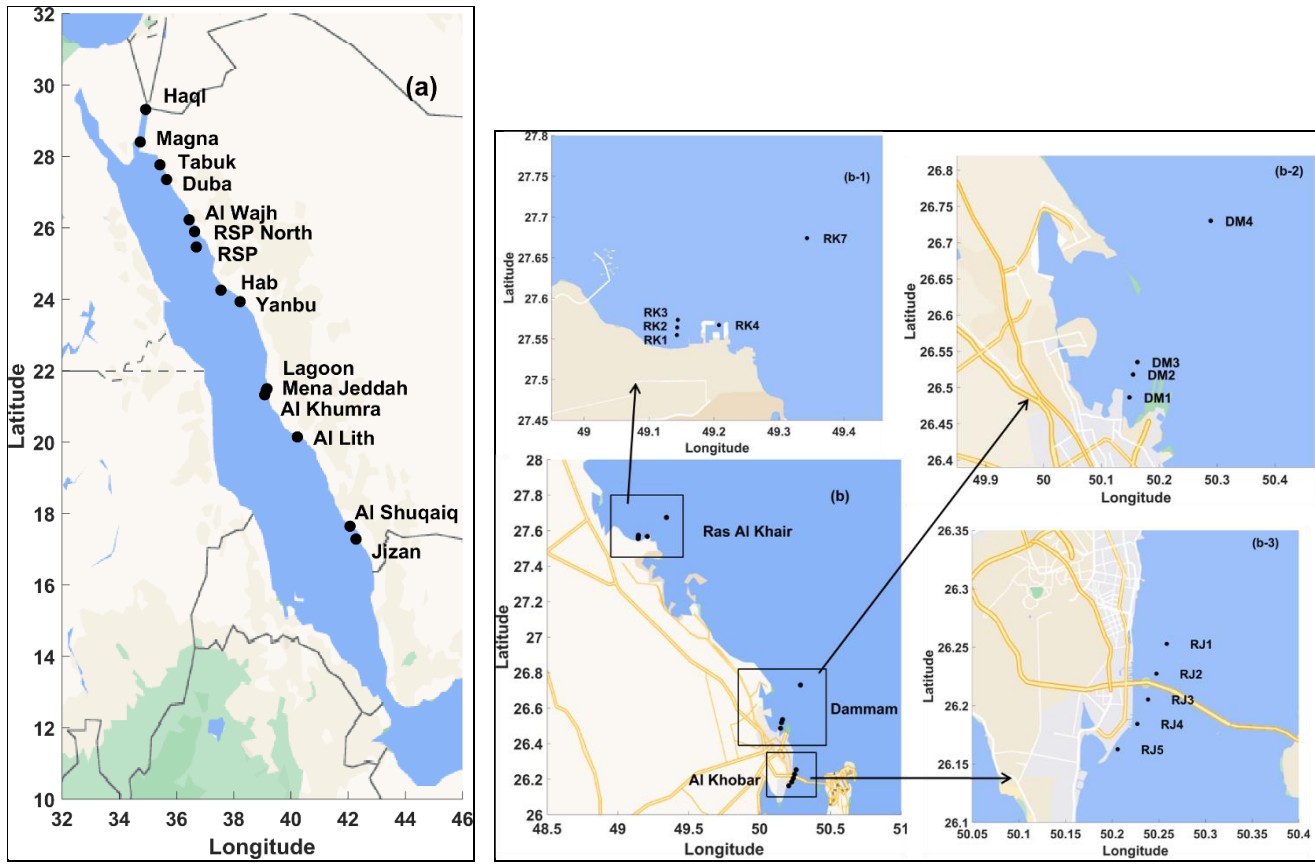

**Figure 1:** Station maps. (a) Sampling areas within the coastal zone of the Red Sea (© Google Maps, 2021). A number of sampling stations were located in each area, as shown in Table 1a. (b) Sampling areas in the Arabian Gulf. Detailed station grid for each area is shown in Fig. 1b-1, 1b-2 and 1b-3.

## 3. Methods

### 3.1. *CTD - Rosette*

In the Red Sea, 96 CTD casts were performed. A total of 71 casts included Rosette/Niskin water sampling (full details are given in Table 1a and 1b) while the other 25 acquired only sensor data. A Seabird SBE 9plus CTD underwater unit (Sea-Bird Electronics, Inc.) connected to an SBE 11 deck unit was used. The

SBE 9plus CTD system was equipped with a pressure sensor (Digiquartz), temperature sensor (SBE 3), conductivity sensor (SBE 4), DO (SBE 43), transmissometer sensor (Chelsea AlphaTracka Mk II), fluorometer sensor (Chelsea AquaTracka III), pH sensor (SBE 18) and underwater and shipborne photosynthetically active radiation (PAR) sensor (Satlantic). Temperature and conductivity sensors were calibrated before the cruise, in February/March 2021, at the manufacturers' facilities. Oxygen values were

corrected post-cruise using the Winkler oxygen values. The underwater unit was attached to a metallic frame that supported 12 Niskin bottles of 12 L volume each. The bottles were connected to a carousel water sampler (SBE 32) linked to the CTD probe via a conductive cable.

In addition, a portable SBE 19 CTD underwater unit equipped with a pressure sensor (strain gauge), unit-embedded temperature sensor, unit-embedded conductivity sensor, transmissometer sensor

(Chelsea/Seatech) and fluorometer sensor (WET Labs ECO-AFL/FL) was used in the shallow areas. Water samples for the Rosette system were taken for the Quality Control/ Quality Assurance (QC/QA) of the results derived from the CTD probes. Moreover, an inter-calibration test cast was conducted before sampling, in which both the SBE 9 and SBE 19 probes were attached to a frame and lowered in the water column. Minor adjustments were applied to correct the SBE19 sensors. In total, 36 casts were conducted

with the tender's CTD unit. Raw CTD data were processed following the manufacturer's recommendations and procedures using the SBE Data Processing software.

The physical properties derived from the CTD and companion sensor data provided indicators of pressure signals.  For example, salinity, fluorescence and beam transmission were taken as tracers for brine water (e.g., from desalination plants), wastewater effluent (e.g., from effluent outfalls).

In the Arabian Gulf, data were acquired with a portable Idronaut Ocean Seven CTD unit equipped with pressure, temperature, conductivity, DO, pH and turbidity sensors. All sensors attached to the unit were calibrated before the cruise, in September 2021, at the Coastal and Marine Resources (CMR) Core Lab of KAUST according to the manufacturer's instructions. Arabian Gulf water samples were taken using Niskin bottles.




### *3.2. ADCP*

The R/V *AEGAEO* was equipped with a portable ADCP operating at 300 kHz. The ADCP (Teledyne Workhorse) measured water current magnitude and direction down to a depth of approximately 50 m in
optimal conditions. The instrument was deployed on the side of the *AEGAEO* at shallow stations with the bottom tracking mode enabled. This ADCP was typically set up to measure velocities in 2-m depth bins, collecting 4–5 (ensemble) profiles at 6–8 min intervals. The average velocity profile at each station was computed as a mean of these ensembles with questionable-quality (biased and noisy) measurements removed. At 15 deeper stations where depths ranged from ~70 to ~600 m, ADCP measurements were
collected for 10–15 min via the shipboard ADCP (SADCP Teledyne Ocean Surveyor), a unit installed at the ship's hull and operating at 75 kHz. Averaged velocity profiles were obtained at a total of 59 stations. Unfortunately, ADCP measurements were not taken in the Arabian Gulf.

### *3.3. DO, Nitrogen and Phosphorus, Chl-a, Total Organic Carbon (TOC) and Total Suspended Solids (TSS)*

Seawater samples were collected from the Rosette/Niskin sampling system with the typical precaution to prevent any biological activity and gas exchange with the atmosphere (Strickland and Parsons, 1968). Winkler glass bottles with bevelled glass stoppers and a measured capacity specification of ±0.01 mL were used. Chemical reagents were added immediately after sampling. DO was determined by titration using the Winkler method, according to Carpenter (1965a,b). Reagent 1 (manganese) and reagent 2
(alkaline iodide) were added with semi-automated Eppendorf (Germany) dispensers, and the bottles were kept in the dark. Before titration, the precipitated hydroxides were dissolved with sulphuric acid, and the solution was carefully transferred to a titration beaker. The titration was carried out with a standardised thiosulphate solution using a Metrohm 876 Dosimat (Switzerland). QC/QA was performed daily by standardisation of the thiosulphate solution with a reference standard solution of potassium iodide. The
precision of the method outlined above is estimated to be 2.2 µmol/L.

The onboard determination of DO concentration from the seawater samples provided data for QC/QA of the CTD DO sensor measurements and enabled calibration of the CTD's DO sensor (SBE 43) during the cruise.

For nutrient analysis, seawater samples were collected in tubes amenable to the QuAAtro nutrient

autoanalyser (SEAL Analytical) and were also collected in 125 mL bottles. All sample containers were
pre-treated with 10% HCl and distilled water. The samples were collected in triplicate and kept deep-
frozen (−20 °C) until their subsequent analysis in the HCMR biogeochemical laboratory, which is
certified according to *EN ISO/IEC 17025:2017*. In the laboratory, the samples were analysed using the
QuAAtro nutrient autoanalyser (SEAL Analytical) according to standard methods (Strickland and

Parsons, 1968; Murphy and Riley, 1962). Ammonium was determined based on the spectrophotometric
measurement of the blue-coloured indophenol complex formed by the reaction of phenol and hypochlorite
in the presence of the $NH_4^+$ and $NH_3$ species (Koroleff, 1970). Absorbance was measured using a UV-
2600 Shimadzu UV/Visible spectrometer (USA) with an 8 mL volume, 10 cm path length cell. QC
samples were analysed together with the field samples. Results were calculated according to a calibration

curve based on the *EN ISO/IEC 17025:2017* standard, followed by the HCMR Laboratory QUASIMEME
inter-calibration exercise used for testing the accuracy of the method. The LOD values of the certified
methods were 0.04 μmol/L for nitrate+nitrite, 0.01 μmol/L for nitrite, 0.03 μmol/L for nitrate, 0.01 μmol/L
for phosphate and 0.02 μmol/L for ammonium.

For organic phosphorus and nitrogen analysis, seawater was collected in 20 mL Teflon bottles (for

phosphorus) and 50 mL glass bottles fitted with screw caps. These digestion bottles were previously
washed with 10% HCl. Preliminary digestion was performed to remove traces of organic matter. The
bottles were then kept continuously in a deep freeze (−20 °C) in the dark until their contents were analysed
in the HCMR laboratory. In the laboratory, digestion was performed by a persulphate wet-oxidation in
low alkaline conditions at 120 °C (1 bar) for 30 min. After cooling at room temperature, the assay mixture

was analysed for nitrate and phosphate on a QuAAtro nutrient autoanalyser (Seal Analytical) according
to the methods for nutrient analysis.

Samples for TOC analysis were collected in 20 mL acid-cleaned (10% HCl, 12 h) glass bottles fitted with
Teflon cups. Directly after sampling, 50 μL of 2N HCl was added to each bottle. The samples were then
refrigerated until analysis. TOC concentrations were determined using a Shimadzu TOC-L organic carbon

analyser following the high-temperature catalytic oxidation (HTCO) method described by Cauwet (1994)

and Sugimura and Suzuki (1988). TOC concentration was calculated as the average value of three replicates that yielded a standard deviation <2%. Analytical precision and accuracy were tested daily against Deep Atlantic Seawater Reference Material provided by the DOC-CRM program (University of Miami—D.A. Hansell). The certified value of the reference material is 0.533–0.574 mg/L, and the measured values (n = 10) during the analysis of the samples were between 0.532 and 0.570 mg/L.

Water samples for the determination of Chl-a concentrations (μg/L) were collected by Niskin bottles principally near the surface of the water column and at the depth of the deep chlorophyll maximum (DCM) concentrations. For the estimation of the phytoplankton biomass, seawater samples were filtered on board through Whatman GF/F microfiber filters.  A volume of 1 or 2 L of seawater were filtered depending on the expected concentrations of Chl-a. The filters were kept in a deep freezer in the dark at –15 °C, and were than analysed at the laboratory on a TURNER 00-AU-10 fluorometer.

TSS was determined by a standard method in which 4 L of seawater was collected in polyethylene bottles and filtered on board through pre-weighed glass fibre filters.

### 3.4. _BOD, Sulphide, Fluorine, Cyanide, Total Chlorine in Seawater_

The determination of BOD was performed on board immediately after seawater sampling according to Standard Methods for the Examination of Water and Wastewater No. 5210b. Seawater samples were taken from the Niskin bottles with the recommended precaution to prevent any biological activity and gas exchange with the atmosphere (Strickland and Parsons, 1968). A WTW 208262 Respirometric BOD Measuring System was used for the analysis. The bottles were incubated at 20 °C for 5 days. The accuracy of the BOD method is ±1 (± 3.55 hPa). The determination of sulphide concentration was performed on board immediately after seawater sampling. Seawater samples were taken from the Niskin sampling bottles, and sulphides were determined by using the photometric method of Cline (1969). Sulphides react with _N,N_-diethyl-_p_-phenylenediamine (DPD) sulphate and ferric chloride to produce a blue colour measured with a UV/Visible spectrophotometer at 670 nm. The limit of detection (LOD) was 1 μg/L.

For fluoride measurement, seawater samples were collected from the Niskin bottles in the special tubes equipped for the ECION700/40S Eutech Ion 700 Meter with Integral Electrode Holder & 100/240 VAC

Adapter CE equipped with Eutech 9609BNWP Fluoride Ionplus Sure-Flow Combination Electrode with a BNC connector.

For cyanide analisis, sea water samples were collected from the Niskin bottles in 125 ml polyethylene bottles. The samples were preserved by adding NaOH until pH > 12 and stored at 4 oC until their analysis at HCMR labs. The analytical methodology was based on the colorimetric method No 4500-CN- E in Standard methods for the examination of water and wastewater. The detection limit is 0.2 µg CN/L.

Chlorine concentration was measured on board immediately (in exactly 2 min) after seawater sampling. Seawater samples were taken from the Niskin bottles, and chlorine was determined by photometry according to the Standard Methods for the Examination of Water and Wastewater No. 4500-Cl G. Chlorine reacts with DPD to produce a red colour measured with a UV/Visible spectrometer at 515 nm. The LOD of the method was 10 µg/L.

### 3.5. _Total Petroleum Hydrocarbons (TPH), Oil/Grease and Chlorophenols in Seawater_

Total petroleum hydrocarbons (TPH), oil, grease and chlorophenols were analysed in the HCMR organic chemistry laboratory certified according to EN ISO/IEC 17025:2017. For TPH, oil and grease determination, 2.5 L of seawater was collected in glass bottles and, after the addition of deuterated _n_-C24 as an internal standard, was immediately extracted on board with _n_-hexane. For chlorophenols, 1 L of seawater was collected in high-density polyethylene bottles, 2-chlorophenol-3,4,5,6-D4 was added as an internal standard, and then the samples were acidified to pH < 4 and extracted on board with dichloromethane. All extracts were stored in a refrigerator and transferred to the HCMR laboratory.

In the laboratory, TPH was determined by gas chromatography–mass spectrometry-FID (Agilent 7890A/5975C GC-MS) based on the ISO 9377-2:2000 method, and oil/grease was determined gravimetrically as the hexane extractable material using EPA method 1664. The hexane was distilled to dryness, and after further drying in an oven at 60 °C, the remaining residue was weighed in a 4-digit analytical balance. The LOD was 0.1 mg/L.



The analysis of chlorophenols was based on EPA method 528. In the laboratory, the extracts were dried with sodium sulphate, and their volume was reduced to about 1 mL using a rotary evaporator and finally to 100 µL using a stream of ultraclean nitrogen. Chlorophenols were determined by gas chromatography–mass spectrometry (Agilent 7890A GC - 5975C MS). The following substances were quantified using the internal standard method: 2-chlorophenol, 3-chlorophenol, 4-chlorophenol, 2,3-dichlorophenol, 2,4-dichlorophenol, 2,5-dichlorophenol, 2,6-dichlorophenol, 3,4-dichlorophenol, 3,5-dichlorophenol, 2,3,4-trichlorophenol, 2,3,5-trichlorophenol, 2,3,6-trichlorophenol, 2,4,5-trichlorophenol, 2,4,6-trichlorophenol, 3,4,5-trichlorophenol, 2,3,4,5-tetrachlorophenol, 2,3,4,6-tetrachlorophenol, 2,3,5,6-tetrachlorophenol and pentachlorophenol. The LOD for each compound was 0.01 µg/L.

### 3.6. *Metals in Seawater*

For the analysis of dissolved Cd, Co, Cr, Cu, Ni, Pb and Zn, 250 mL of seawater was collected in pre-cleaned polyethylene bottles and stored at $-20$ °C until analysis in the laboratory. After thawing, the samples were filtered through 0.45 µm filters (Whatman sterile mixed cellulose ester membranes) and acidified with Suprapur HCl to pH < 2. Samples for Cd, Co, Cu, Ni, Pb and Zn determination were pre-concentrated by the Toyopearl AF Chelate 650M resin to separate these elements from interfering matrix components (Milne et al., 2010; Willie et al., 1998). The trace metals eluted on the resin were collected with 1 M Suprapur $HNO_3$ and determined using inductively coupled plasma mass spectrometry (ICPMS, Thermo-Elemental X-series II) in a regular laboratory environment. Accuracy and precision were assessed using the Cass-5 certified reference material for coastal water and acidified seawater samples of the QUASIMEME inter-laboratory exercise (AQ-3, Lab code 122, www.quasimeme.org). The results obtained for Cass-5 were in good agreement with the certified values (Tables S1 and S2), and the samples of the QUASIMEME tests had acceptable Z-scores ($-2 < Z < 2$).

A co-precipitation method was used for total dissolved Cr (Harper and Riley, 1985). The samples were precipitated with Fe(II) ammonium sulphate. Total dissolved Cr determination was performed by graphite furnace atomic absorption spectrometry (AAS, Shimadzu GFA-7000A) in a regular laboratory environment. The LOD of the method was 0.083 µg/L.

For the total Hg, 250 mL of seawater was collected in pre-cleaned glass bottles. After adding 1.2 mL of 37% HCl, the samples were stored in refrigeration until the analysis in the laboratory. Total Hg was determined by EPA method No 1631, which consisted of oxidation of all species to Hg(II), purging and trapping onto a gold trap, desorption and cold-vapor atomic fluorescence spectrometry (CVAFS) by a Tekran 2500 mercury analyser. The LOD was 0.2 ng/L, and the limit of quantification (LOQ) was 0.5 ng/L. On each day of analysis, the reference material was analysed at least once daily for recovery estimation (accuracy) (Tables S1 and S2). The reference material was a diluted sample from digested sediment with a certified Hg content of 412 μg/kg, an aliquot of which was spiked in purged seawater to reach a concentration of 2.0 ng/L. Thus, both accuracy and matrix spike checks were performed.

The samples' pre-treatment, the trace metal pre-concentration steps, the trace metal analyses and Hg determination were carried out in U.S. FED-STD Class M5.5 (10,000) cleanroom environments using ultraclean handling techniques (EPA method 1669).

### 3.7. *Sediments*

Surface sediment samples were collected for the determination, using EPA methods, of aliphatic and polycyclic aromatic hydrocarbons (AHC and PAH), volatile organic compounds (VOC) benzene, toluene, ethylbenzene and xylene (BTEX), polychlorinated biphenyls (PCB), phenols, organic carbon and metals. Organic compounds and metals were analysed in the HCMR organic chemistry laboratories certified according to EN ISO/IEC 17025:2017. An internal quality check was performed by means of analyses of QUASIMEME samples. In order to identify the geochemical background of each area, additional sampling and analysis of sediment cores is needed.

The analysis of AHC and PAH in sediments was based on the recommendations of OSPAR (2013) and UNEP/IOC/IAEA (1992). The sediments were frozen at −20 °C (UNEP, 1992) until they arrived at the laboratory. In the laboratory, the sediment samples were dried at 40 °C, sieved through a 250 mm sieve and homogenised. After the addition of a mixture of deuterated compounds, used as internal standards, 0.5–3 g of dried sediment was extracted with a mixture of methanol and dichloromethane (1:2, v/v) using an accelerated solvent extractor (Dionex ASE 350). The extract was saponified with methanolic KOH.

and the unsaponified components were extracted with *n*-hexane and cleaned up and fractionated by passing through a silica column. The final determination of total AHC and PAH was carried out by gas

chromatography−mass spectrometry (Agilent 7890A/5975C GC-MS). The quantitation was based on the deuterated internal standards. Blanks were systematically checked to verify the absence of contamination during analyses. Accuracy ranged from 79.8% to 96.5% for individual PAH compounds and was systematically controlled using a reference material (NIST SRM 1941b), and the laboratory also participates in QUASIMEME inter-laboratory exercises. Analytical uncertainties for each PAH ranged

from 12.1% to 37%, $k = 2$. The LOD for total AHC and PAH was 0.5 µg/g and 0.1 ng/g, respectively. Individual PAHs determined were naphthalene, acenaphthylene, acenaphthene, fluorene, dibenzothiophene, phenanthrene, anthracene, fluoranthene, pyrene, benzo(*a*)anthracene, chrysene, benzo(*b*)fluoranthene, benzo(*k*)fluoranthene, benzo(*e*)pyrene, benzo(*a*)pyrene, perylene, indeno(1,2,3-*cd*)pyrene, benzo(*ghi*)perylene, dibenzo(*ah*)anthracene and the methylated derivatives of naphthalene,

dibenzothiophene, phenanthrene, pyrene and chrysene.

For the analysis of PCB in sediments, 3 g of dried sediment were extracted, following the addition of a mixture of CB112, CB155 and CB209 (used as internal standards), with a mixture of hexane and dichloromethane (1:1, v/v) using an accelerated solvent extractor (Dionex ASE 350). The extract was cleaned on an alumina glass column. The final determination of PCB was carried out by capillary gas

chromatography using a non-polar column and an electron capture detector (Agilent 7890A GC). The quantitation was based on the abovementioned internal standards. Accuracy ranged from 69.3% to 83.5% for each congener and was controlled using a reference material (NIST SRM 1941b). Analytical uncertainties for each congener ranged from 27.1% to 39%, $k = 2$. The following individual congeners were determined: CB28, CB52, CB101, CB118, CB105, CB138, CB153, CB128, CB156, CB170,

CB180, CB183 and CB194. The LOD for each congener was 0.01 ng/g.

The analysis of BTEX was performed using the equilibrium-based static headspace technique and gas chromatography−mass spectrometry according to EPA methods 5021A and 8260. Briefly, ~3 g of wet sediment was collected in 20 mL headspace vials and kept at −20 °C until arrival at the HCMR laboratory. In the laboratory, VOC was determined using a headspace autosampler (HTA, HT2800T) and a gas

chromatography–mass spectrometry system (Shimadzu GCMS-QP2020 NX). The following substances were quantified using an external standard mixture: benzene, toluene, ethylbenzene, *o*-, *m*- and *p*-xylenes. The LOD was 0.05 μg/kg.

The analysis of phenols in sediments was based on the EPA (spectrophotometric) method 420.1. The sediments were frozen at -20 °C until their arrival in HCMR labs. In the lab, 15 g of sediment were put in

500 mL of distilled water and pH was adjusted <4 using phosphoric acid. The sample was distilled and total phenols were determined in the distillate by the 4-aminoantipyrine colorimetric method. The dye was extracted with chloroform and absorbance was measured at 460 nm using a UV-2600 Shimadzu UV/Visible Spectrometer and 100mm cell. The detection limit was 0.1 μg/g

Major elements and trace metals in marine sediments were measured by an accredited method (EN

ISO/IEC 17025:2017) using a wavelength-dispersive X-ray fluorescence (WDXRF) system. The LOD for this XRF method was calculated from a series of measurements of 5 samples for major elements and 10 samples for trace elements using the certified reference sample PACS-2. The LOD measurements were carried out in the same experimental conditions in which the sediment samples were analysed (Table S3).

The calibration of the XRF method was carried out by scanning reference samples that contained a wide spectrum of element concentrations. For this calibration analysis, several rock and sediment samples were gathered, mainly from the U.S. Geological Survey and the National Research Council of Canada Reference Materials. Fused beads and powder pellets were prepared carefully, and all samples were scanned for major and trace elements. All measurement parameters were configured through the software,

building two distinct "applications" and selecting the optimum settings for each element separately. Subsequently, element concentrations were plotted against the measured intensities, and a linear fit was generated by regression. Theoretical alpha corrections and possible line overlaps were carefully resolved until the lowest mean error of the fit was obtained. Accredited trace elements are As, Co, Cr, Cu, Mn, Mo, Ni, Pb, Sb, Sn, Sr, V, Zn, Ag, Ba, Bi, Br, Cd, Ce, Cs, Ga, Ge, Hf, Hr, I, La, Nb, Nd, Rb, Sb, Sc, Se,

Sm, Sn, Sr, Ta, Te, Th, Tl, U, W, Y, Yb and Zr. Accredited major elements are $Fe_2O_3$, $CaO$, $TiO_2$, $Al_2O_3$, $K_2O$, $MgO$, $Na_2O$, $P_2O_5$, $SO_3$ and $Si_2O$.



## 4. Results and Discussion

Here, we present the essential physical and biochemical parameters related to the hydrography and the quality of seawater and sediments of the Arabian Gulf and Red Sea study areas (SEANOE. https://doi.org/10.17882/96463). A principal goal is to trace contaminants from pollution sources along the coastal zone of Saudi Arabia. In addition, the data establish the baseline for the future design of monitoring strategies of the Saudi Arabian coastal marine environment.

### 4.1. Red Sea

#### 4.1.1. Physical Parameters

The water properties measured during the Red Sea cruise would be expected to conform with the general picture of the regional physical oceanography prevailing during the sampling period.

The temperature–salinity (T–S) diagram presenting all the CTD casts of the Red Sea surveys displays the general physical characteristics of the water masses (Fig. 2). Expectedly, general gradients in both temperature and salinity are evident, with temperature increasing and salinity decreasing from north to south, a trend consistent with the previous studies of Neumann and McGill (1962), Maillard and Soliman (1986), Sofianos and Johns (2007) and Ali et al. (2018). In particular, the densest water, with the highest salinity and lowest temperatures, is found within the Gulf of Aqaba. In general, the potential temperature (hereafter, temperature) values ranged between 21 and 36 °C which are typical for mid-summer period (Sofianos and Johns, 2008). The north-south gradient of surface salinity, typical for the Red Sea due to the general northward propagation of relatively fresher water from the Gulf of Aden (Sofianos and Johns, 2007; Churchill et al., 2014), is evident, with high surface values (40.7) within the Gulf of Aqaba, decreasing to 38 towards the southern region near Jizan area. The CTD profiles acquired at shallow stations (<100 m) show almost constant temperature and salinity with depth; whereas, at the deeper areas, the temperature tends to decrease while salinity increases with depth. Additionally, due to the shallow bathymetry of the coastal areas, they are more susceptible to local atmospheric forcing, with increased temperature compared to the offshore during the cruise period.

The T-S properties at the hotspot sites reveal potential signs of pressures on the coastal ecosystem. Low salinity, detected within the Jeddah lagoon system, indicates the input of fresher water, possibly consisting of sewage effluent as observed by Garcia et al. (2014). A slight increase in temperature and salinity within

the Islamic Port of Jeddah compared to the stations located outside the port, could be attributed to water stagnation in the port. Differences in temperature and salinity values at Al Shuqaiq and Al Lith, can be attributed to water discharges linked to desalination plant and aquaculture activities, respectively.

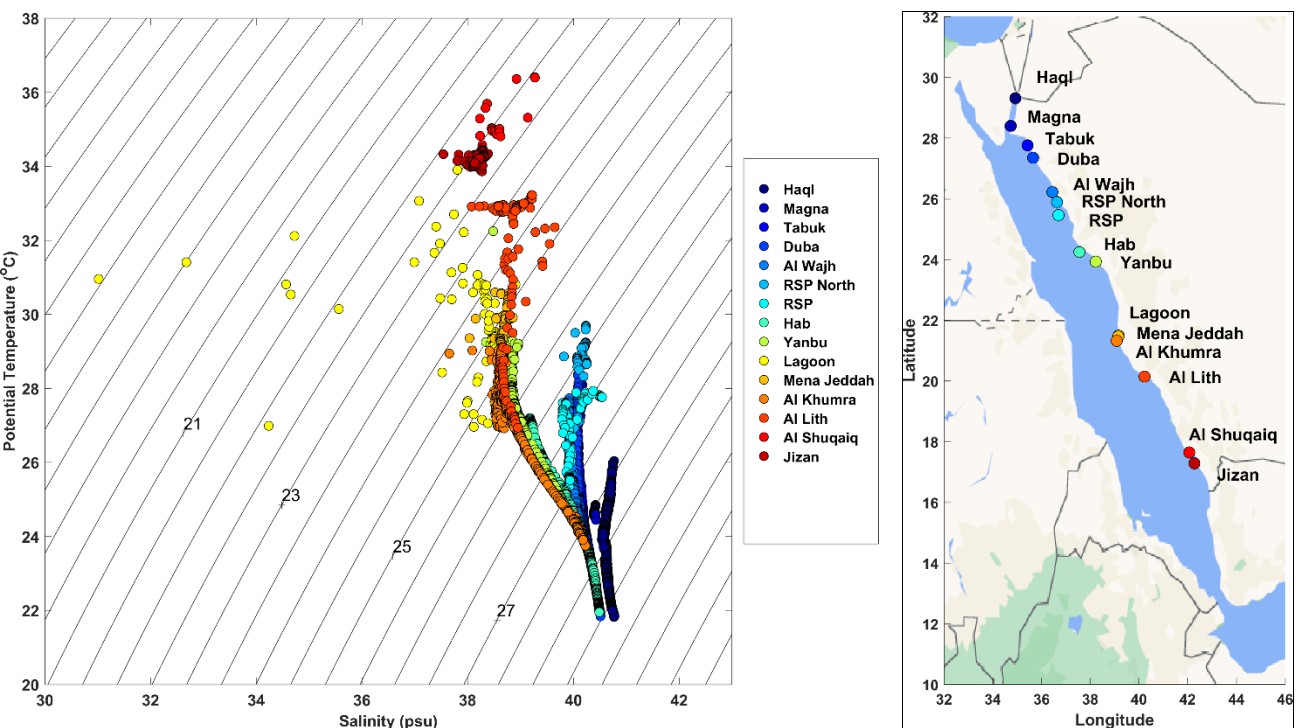

**Figure 2:** T–S diagram encompassing all data taken in the Red Sea. Dots of the same colour indicate measurements taken at

the same site as shown on the map on the right (© Google Maps, 2021).


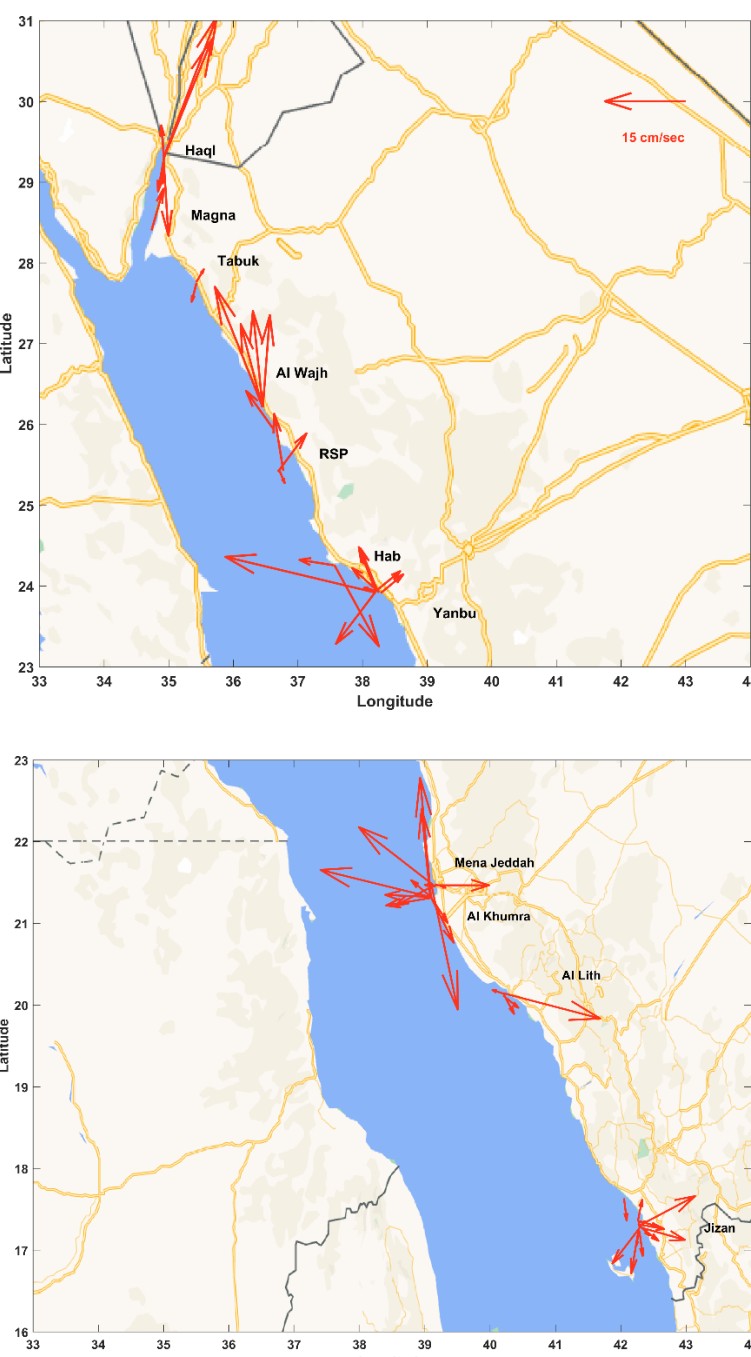

**Figure 3:** Current vectors at 12 m at specific areas along the Saudi Arabian coast of Aqaba and Red Sea. Scale vector of a westward current with a speed of 15 cm/s appears in the upper right of the top panel (© Google Maps, 2021).


Regarding the current measurements, the surface currents range from weak, on the order of ~2−3 cm/s, to substantial, reaching values of 17−25 cm/s. Dominant northward flow only appears at stations located in the north Red Sea, whereas, multi-directional flow seems to be the rule at most stations (Fig. 3). Pavlidou et al. (2021) have found that sub-surface currents in the depth range of 18−60 m are mostly in

the same direction as the near-surface ones and parallel to the nearby coast.

### *4.1.2.   Biogeochemical Parameters*

The results obtained during the cruise in June 2021 in the Red Sea coastal area, showed that the Gulf of Aqaba appears to be better oxygenated through the whole water column, from the surface to the bottom layer, while a decrease in oxygen near the bottom (115 μmol/L), which is possibly connected to the

organic material accumulated near the bottom, is recorded at the majority of the sampling stations. As recently reported by Povinec et al. (2023), in the Gulf of Aqaba, the first 300 m of the water column indicate stable DO values of about 210 μmol/L, whereas DO of about 3.5 μmol/kg was measured below the depth of 500 m. Conversely, in the open Red Sea, DO minimum values from 0.6 to 1.25 μmol/kg were observed at depths from 300 to 450 m, showing a decreasing trend below 150−200 m.

The assessment presented herein aims to identify the areas facing eutrophication and/or pollution problems. In general, in the Red Sea, north−south increasing gradients were evident for some of the parameters studied, revealing a link between the hydrographic conditions and biogeochemical properties. Nutrients and organic carbon revealed high values at Jeddah Lagoon (Figs. 4 -6), which is a unique system with low water renewal and pollution mainly from domestic sewage ((Peña-García et al., 2014). The

nitrite values at Jeddah Lagoon were high (3.06-3.90 μmol/L) and the highest ammonium values (2.90 μmol/L) were found at the station (L03), whereas total dissolved nitrogen was extremely high at the same station, reaching 167 μmol/L. At all other sites of the Red Sea, nitrate + nitrite was low in the euphotic zone, and increased with depth. Nitrate + nitrite values in the euphotic layer (approximately 0-100 m) ranged between below the LOQ and 2.30 μmol/L, with the exception of two relatively high values

detected at Al Khumra, close to the sewage outfall (3.43 μmol/L), and station AL1 of the Al Lith grid

(5.78 μmol/L). These values exceed the commonly observed values in coastal areas and could indicate organic load from anthropogenic activity. Figs. 7 and 8 illustrate the distribution of Chl-a and selected pollutants (metals and organic) along the north–south section. Higher Chl-a values recorded in the southern Red Sea correlate with relatively higher nutrient concentrations, which are influenced by the

Gulf of Aden Intermediate Water (GAIW); the inflow of nutrient-rich water entering the Red Sea from the Gulf of Aden through the Strait of the Bab-el-Mandeb (Churchill et al., 2014). In this survey, the Chl-a concentrations did not exceed 1 μg/L at any site except from Jeddah Lagoon, where extremely high Chl-a values reaching 40 μg/L were sometimes measured within the Jeddah Lagoon System. The Kingdom of Saudi Arabia has set maximum allowable values only for ammonium (5.5 μmol/L), whereas it has not set

maximum allowable values for Chl-a. The Abu Dhabi Quality and Conformity Council (2018) set 1 mg/L as the maximum allowable concentration of Chl-a for ambient marine waters, whereas in the Eastern Mediterranean Sea the target values for the Good Environmental status is 0.53 μg/L (EC Decision 2018/229/EU). Regarding Chl-a distribution, it seems that the ecological status of the water at the southernmost site of the Red Sea, Jizan Economic City, is classified as poor, which corroborates the algal

blooms observed in this area during summer and early Autumn 2021 (personal communications with the Emirate of Jazan Province). This area may be affected by water intrusion from the south and atmospheric deposition. The deposition of material transferred by dust storm events may also influence the ecological quality of the water in the area, since during the summer period, the Tokar Gap frequently channels strong winds onto the sea surface, causing African dust storms spreading over the southern part of the Red Sea

(Jiang et al., 2009; Garrisonet al., 2010).

Sediments at all of the Red Sea study sites are found to be enriched in some metals (e.g., As). This finding should be further investigated via sampling and analysis of sediment cores in order to define the geochemical background of the region. Moreover, AHC, PAH and PCB, which constitute important classes of organic contaminants that may cause degradation and pose a risk of serious damage in the

marine environment, are determined in surface sediments collected from the coastal zone of the Red Sea (Fig. 9). The examination of various indices reveals a chronic petroleum-associated anthropogenic pressure in Jeddah Lagoon System, Jeddah Sea Port (Mena Jeddah) and Al Khumrah, whereas some



petroleum residues were also found at King Fahd Yanbu Port, Mena Jeddah, shrimp and fish farms near Al Lith and to a lesser extent at Magna.

Organic pollution in the Jeddah Lagoon System and at King Fahd Port of Yanbu is also confirmed by BOD values (4.5 mg/L inside the port to 7.7 mg/L in the lagoon), and fluoride values at the northern part of the Gulf of Aqaba confirmed the effect of the phosphate terminal in the Port of Aqaba in Jordan due to cross-border pollution. However, it seems that industrial activities probably enrich the coastal zone of the Red Sea with organic pollutants.






**Figure 4:** Average ammonium (left panel) and phosphate (right panel) concentrations in the sampled marine areas along the coastline of the Red Sea (© Google Maps, 2021). The mapping presents the average concentrations for each area (average from all stations sampled in the area, average of mean integrated concentrations of the stations included in each area). Ammonium values at Jeddah Lagoon are out of scale.


**Figure 5:** Average Total Dissolved Nitrogen (left panel) and organic carbon (right panel) concentrations in the sampled marine areas along the coastline of the Red Sea (© Google Maps, 2021). The mapping presents the average concentrations for each area (average from all stations sampled in the area, average of mean integrated concentrations of the stations included in each area). Jeddah Lagoon values are out of scale, since they ranged from 11.8 to 167 µmol/L for total dissolved nitrogen and from 595 1 to 5.7 mg/L for total organic carbon.

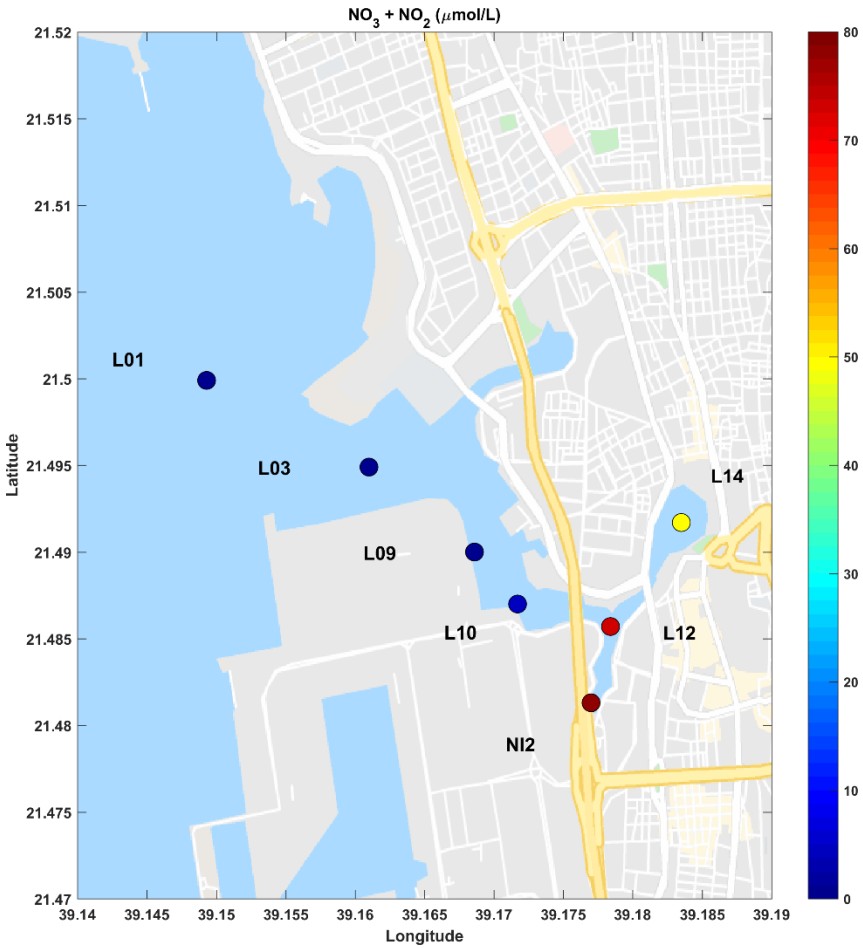

**Figure 6:** Nitrate + Nitrite concentrations in Jeddah Lagoon (© Google Maps, 2021) .





**Figure 7:** Average Chl-a concentrations in the sampled marine areas along the coastline of the Red Sea (left panel) and the indicative assessment of ecological status based on Chl-a concentrations according to the five-scale classification scheme for the Eastern Mediterranean (right panel). The mapping presents the average concentrations for each area (average from all stations sampled in the area, average of mean integrated concentrations of the stations included in each area) Chl-a values at Jeddah Lagoon (left panel) ranged from 0.61 to 44 µg/L (© Google Maps, 2021).



**Figure 8:** Average total mercury (Hg; ng/L) (left panel) and petroleum hydrocarbon (TPH) concentrations (µg/L) (right panel) at the stations sampled in the Rea Sea (© Google Maps, 2021). The mapping presents the average concentrations for each area (average from all stations sampled in the area, average of mean integrated concentrations of the stations included in each area).

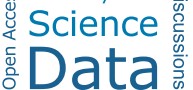





**Figure 9:** Total aliphatic hydrocarbon concentrations (TAHC; mg/kg), total EPA PAH concentrations ($\sum PAH_{EPA}$), U/R diagnostic ratio and percentage contribution of pyrolytic PAH in sediments for each site in the Red Sea (© Google Maps, 2021). The mapping presents the average concentrations for each area (average from all stations sampled in the area, average of mean integrated concentrations of the stations included in each area). For the U/R diagnostic ratio, the colour scale indicates average values >4 (red colour) or <4 (blue colour) for all study sites.

### *4.2. Arabian Gulf*

#### *4.2.1. Physical Parameters*

Salinity is the main indicator for local differentiations in the Arabian Gulf. A broad range of salinity values is observed, with higher salinity close to the coastline at Ras Al Khair and, especially, at Al Khobar, reflecting the effects of the local desalination plants and their brine discharges. The CTD casts conducted during the survey are plotted in a T–S diagram (Fig. 10) with the different sampling sites distinguished by different colours. In general, temperature values ranged from 31.0 to 33.5 °C. The salinity measurements span a wide range of hypersaline values, from 40 to 52. Along with the hydrological characteristics of the gulf and the regional morphology, the presence of this desalination plant seems to be the major factor that leads to extremely high salinity values at the study site of Al Khobar.

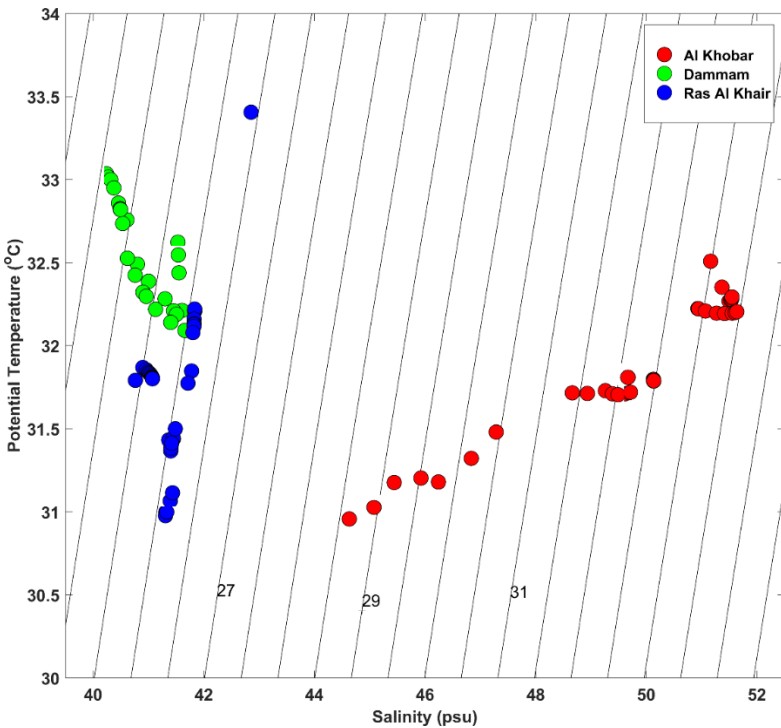

**Figure 10:** T–S diagram for all measured stations in the Arabian Gulf. The three colours of dots represent the three hotspot sites.

### *4.2.2. Biogeochemical Parameters*

In terms of the eutrophication and pollution status of the three areas in the Arabian Gulf, the Dammam is supposed to be affected by wastewater discharges (Mahboob et al., 2023). The concentrations of inorganic nutrients and organic phosphorus are low at all three sites, indicating that eutrophication is not affecting these specific sites during the sampling period. However, dissolved organic nitrogen and carbon values

are relatively high, reaching a maximum value of 18.8 µmol/L for nitrogen and 477 µmol/L for organic carbon. Relatively higher Hg and Pb concentrations (0.55 – 5.90 µg L$^{-1}$ for Hg and 0.18 – 1.25 µg L$^{-1}$ for Pb) were found in the water column (Fig. 11), which is probably linked to the industrial activities in these areas and/or atmospheric deposition (for Pb). However, it should be noted that this analysis only provides a snapshot of the status in the water column at one point in time and that eutrophication-related parameters

exhibit strong seasonal variation.

Similar concentration ranges of metals were detected in the Arabian Gulf and Red Sea coastal waters. It is noteworthy that TPH was low in the Arabian Gulf. However, sediments at Ras Al Khair were found to

be polluted with metals and, in some cases, exceeded the allowable values set in the Abu Dhabi Specifications: 7.0 mg/L for As, 52 mg/L for Cr, 30 mg/l for Pb, 125 mg/L for Zn (Abu Dhabi Quality

and Conformity Council, 2017) (Fig. 12). By contrast, TPH was surprisingly low in the Arabian Gulf. However, sediments at Ras Al Khair, and especially at Al Khobar, were found to be polluted with metals. Regarding the degree of contamination, it seems that sediments in the Arabian Gulf are severe-to-heavily (Al Khobar) metal-polluted. Thus, an additional, more detailed study of the Arabian Gulf, with finer coverage of the coastal zone from Khafji to Al Khobar, is highly recommended.

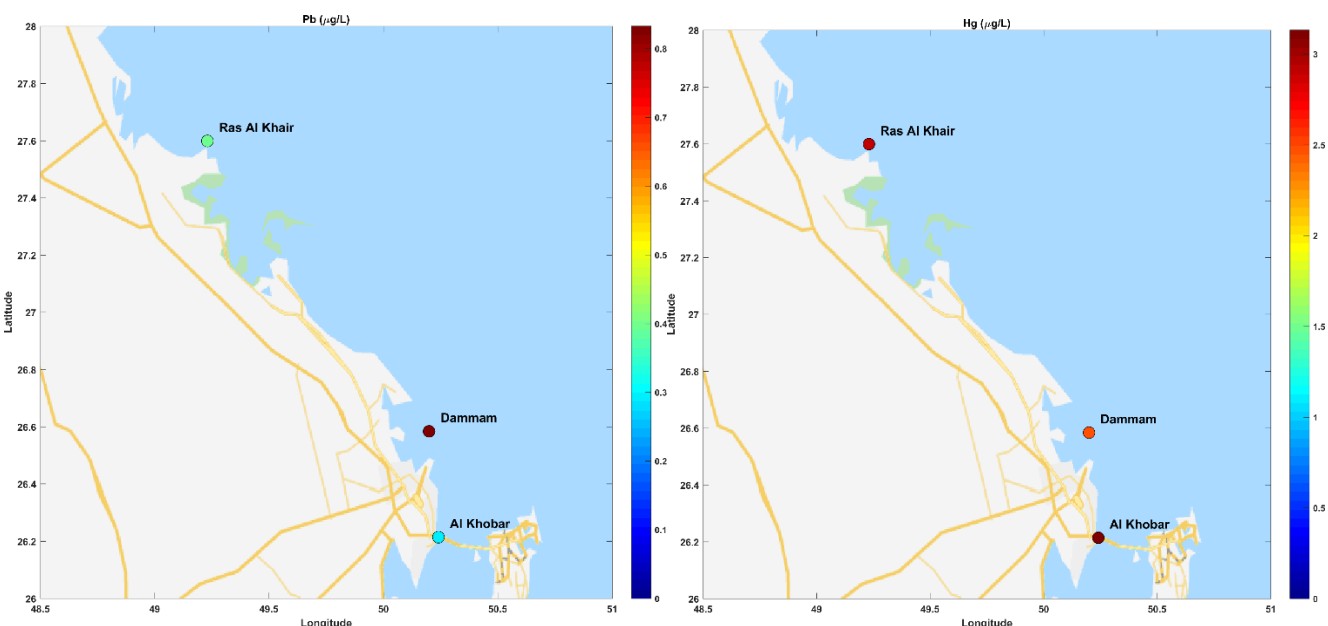


**Figure 11:** Average lead (Pb; mg/L) (left panel) and total mercury (Hg; ng/L) (right panel) in the Arabian Gulf (© Google Maps, 2021). The mapping presents the average concentrations for each area (average from all stations sampled in the area, average of mean integrated concentrations of the stations included in each area).

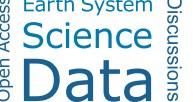


**Figure 12:** Average metals in sediments in the Arabian Gulf (© Google Maps, 2021). The mapping presents the average concentrations for each area (average from all stations sampled in the area).

## 5. Data Availability

Data described in this manuscript can be assessed at SEANOE. https://doi.org/10.17882/96463
(Abualnaja et al., 2023).

## 6. Conclusions

This this is the first broad coverage study in a one-off sampling campaign in Saudi Arabian coastal zone. To the best of our knowledge, these cruises constitute the first multidisciplinary and geographically comprehensive survey of contaminants within the Saudi Arabian coastal waters and sediments, extending

from near the Saudi-Jordanian border in the north of Red Sea to Al Shuqaiq and Jizan Economic City (close to the Saudi-Yemen border) in the south, and in the Arabian Gulf, includes the areas of Al Khobar, Dammam, and Ras Al Khair. The assessment presented in this work, aimed to identify the areas facing eutrophication or/and pollution problems. In general, in the Red Sea, north-south increasing gradients were evident for some of the parameters studied, revealing a link between the hydrographic conditions

and biogeochemical properties. Sediments at all of the Red Sea study sites were found to be enriched in arsenic. In the Arabian Gulf, salinity was defined as the main indicator of local differentiation. A broad range of salinity values were observed, reflecting the effects of the local desalination plants and their brine discharges.

## Competing Interests

The contacts authors have declared that none of the authors have any competing interests.

## Acknowledgements

This publication is based on work supported by the funded project "Marine and Coastal Assessment Protection Study for the Kingdom of Saudi Arabia Coastline" (under contract number 062/475000300/284) made between the Saudi National Center for Environmental Compliance (NCEC)

and King Abdullah University of Science and Technology (KAUST). Captain Theodoros Kanakaris, the crew and the scientific staff of the R/V *AEGAEO* are acknowledged for their assistance and valuable support during the cruise. We especially appreciate the collaboration with and advice offered by the other



Marine and Coastal Assessment Protection Study Initiative members, who hail from different departments at the NCEC and KAUST.

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
