# Peer review of "The Physical and Biogeochemical Parameters along the Coastal Waters of Saudi Arabia during Field Surveys in Summer, 2021"

_Earth System Science Data, 2023_

## Author Response (AR1)

**Reviewer 1:**

Comment 1

The data set is novel and important, especially for future investigations in the region and in other seasons. The paper is well written and adequate in length.

The data set consists of CTD data from the Red Sea taken at stations along the Saudi Arabian Coast. In Fig. 2a it is evident that depth profiles were taken. Samples analysed for nutrients and geochemical variables evidently consist only of surface samples. There should be a short note in the methods section about this: It looks like depth profiles were taken by the CTD but sampling was carried out only in surface waters. How deep did the CTD go?

*Response to Reviewer's 1 comment 1:*

*We'd like to thank you for your time to review our ms and for your comments. We have revised the ms according to your suggestions.*

*Samples analysed for nutrients and geochemical variables do not consist only of surface samples. In table 1c we have provided the sampling discrete depths, which will be clearly shown in the revised table 1c. Moreover, in L 177-180 of the revised ms it is written: "Water samples were taken at discrete depths, such as surface, 10 m, 20 m and near the bottom (roughly 0.5 m from the seabed), as well as from depths of particular interest (e.g., where effluent plumes were identified) if they did not match with the discrete depths".*

*At the methodology part, L. 262-265 of the revised ms, the information for CTD has been added: "Both CTD sensors recorded measurements in the whole water column from near surface to ~ 0.5 m above the sea bottom".*

*Info regarding the T/S diagram has been added in the legend of Fig.2*

Comment 2

In the data set from the Arabian (Persian) Gulf the question also arises whether CTD data in Figure 10 are from different depths or from different stations. The data set in the repository shows that only three samples from each sampling area were chemically analysed. This should also be noted in the methods to make things clear to the reader/user of the data set. In general, the data are well presented and discussed with the relevant literature. The presentation of Arabian Gulf data is a bit shorter but adequate.

*Response to Reviewer's 1 comment 2:*

*The info in included in the methodology part L. 177-180 and L. 262-265. Moreover, info has been added in the legend of Fig. 10.*

*Nutrients and geochemical variables presented in the figures are average values, as written in the figure's legends.*

Comment 3

However, the font in the Figures needs to be enlarged to be able to read the numbers and also the location names. This refers especially to Figures 9, 11 and 12. Some of the other Figures also have slightly small fonts but they are readable. It would be good to adjust all Figures to similar fonts.

*Response to Reviewer's 1 comment 3:*

*All figures have been re-produced and the fonds have been enlarged in order to read the numbers and also the location names.*

Comment 4

Line 641: a reference to the metal data is missing.

*Response to Reviewer's 1 comment 4:*

*Sentence has been modified as follows: "The measurements showed that similar concentration ranges of metals were detected in the Arabian Gulf and Red Sea coastal waters in June 2021'" as it refers to our data and are not compared to other works. L. 679-680*

**Reviewer 2:**

Comment 1

The multidisciplinary data set here presented is relevant for environmental assessment. It can be a reference for future investigations and to check for the effectiveness of environmental protection policies. Moreover, it is particularly precious as very few data are available for these two regions. The data set is well documented with a wide description of meta-data and the paper is well organised. Bibliography is exhaustive and complete. Below some suggestions and small amendments.

-In the Introduction add some information about tidal regime in the regions; this also help for a better interpretation of ADCP currents, water column mixing and dispersion.

*Response to Reviewer's 2 comment 1:*

*We'd like to thank you for your time to review our ms and for your comments.*

*We have revised the ms according to your suggestions. More specifically, in the introduction part we have added information about the tidal regime in the two regions L.74-78 and L. 98-100 in the revised ms, as well as in the Results and Discussion L 549-552 of the revised ms.*

Comment 2

-Desalinization plants strongly increase local salinity. How wide can be the area affected by this very high salinity?

*Response to Reviewer's 2 comment 2:*

*We have added this information at L. 524-529 and 654-656 for the Red Sea and the Arabian Gulf, respectively.*

**Comment 3**

-Can you report about T and S average values in the open sea of Arabian Gulf, for a better comparison with the coastal observations?

*Response to Reviewer's 2 comment 3:*

*We have added this information at L. 648-651 in the revised ms.*

**Comment 4**

-Show a map with bathymetry

*Response to Reviewer's 2 comment 4:*

*A map of bathymetry is shown, as requested. Fig 1 has been replaced showing the bathymetry of the two areas.*

**Comment 5**

-Add a table with the accuracies of the CTD probes sensors

*Response to Reviewer's 2 comment 5:*

*Table 3 with accuracies of the CTD probes sensors has been added (L. 270-271).*

**Comment 6**

-Tab 1a and 1b -In the first row indicate Date_June 2021

-Tab 1c -Indicate that reported values are the sampling depth

*Response to Reviewer's 2 comment 6:*

*Date June 2021 has been added in Tables 1a and 1b, whereas in Table 1c the reported sampling depth is indicated.*

**Comment 7**

-In addition to TS diagram in Fig.2 also provide 2 figures with vertical profiles of temperature and salinity

*Response to Reviewer's 2 comment 7:*

*Figures 3 and 12 have been added showing the vertical profiles of T and S.*

Comment 8

Fig.8 right (TPH) use a small-range scale e.g. 0-20 to better enhance the differences between stations

*Response to Reviewer's 2 comment 7:*

*Fig 8 has been replaced with a small-range scale as requested.*